# synNotch-programmed iPSC-derived NK cells usurp TIGIT and CD73 activities for glioblastoma therapy

Severe heterogeneity within glioblastoma has spurred the notion that disrupting the interplay between multiple elements on immunosuppression is at the core of meaningful anti-tumor responses. T cell immunoreceptor with Ig and ITIM domains (TIGIT) and its glioblastoma-associated antigen, CD155, form a highly immunosuppressive axis in glioblastoma and other solid tumors, yet targeting of TIGIT, a functionally heterogeneous receptor on tumor-infiltrating immune cells, has largely been ineffective as monotherapy, suggesting that disruption of its inhibitory network might be necessary for measurable responses. It is within this context that we show that the usurpation of the TIGIT − CD155 axis via engineered synNotch-mediated activation of induced pluripotent stem cell-derived natural killer (NK) cells promotes transcription factor-mediated activation of a downstream signaling cascade that results in the controlled, localized blockade of CD73 to disrupt purinergic activity otherwise resulting in the production and accumulation of immunosuppressive extracellular adenosine. Such "decoy" receptor engages CD155 binding to TIGIT, but tilts inhibitory TIGIT/CD155 interactions toward activation via downstream synNotch signaling. Usurping activities of TIGIT and CD73 promotes the function of adoptively transferred NK cells into intracranial patient-derived models of glioblastoma and enhances their natural cytolytic functions against this tumor to result in complete tumor eradication. In addition, targeting both receptors, in turn, reprograms the glioblastoma microenvironment via the recruitment of T cells and the downregulation of M2 macrophages. This study demonstrates that TIGIT/CD155 and CD73 are targetable receptor partners in glioblastoma. Our data show that synNotch-engineered pluripotent stem cell-derived NK cells are not only effective mediators of anti-glioblastoma responses within the setting of CD73 and TIGIT/CD155 co-targeting, but represent a powerful allogeneic treatment option for this tumor.

Despite possessing a formidable innate ability to eliminate foreign targets, natural killer (NK) cells fall prey to dysfunction in the microenvironment of glioblastoma (GBM). As a consequence, immunotherapies and chemotherapies for GBM have had minimal improvements on patient survival in the past several decades[1]. The Checkmate series of trials (143, 548, 498) evaluating nivolumab alone or in combination with other therapies has not, for instance, demonstrated significant clinical benefit in GBM patients[2,3]. While many of the

✉ e-mail: sandro@purdue.edu

pathways of dysfunction have included a shift in the balance of surface receptors toward inhibition via upregulation of receptors such as NKG2A[4] and PD-1/PD-L1[5,6], soluble factors—including metabolites, oncometabolites and immunosuppressive cytokines—also play contributing roles by redirecting the metabolism of NK cells toward one that is impermissible to effective anti-tumor responses[7]. One such axis that contributes to NK cell dysfunction involves the cancer-associated poliovirus receptor CD155 and its cognate ligand TIGIT[8], expressed on NK cells. As a stress-related protein, CD155 has multiple binding partners on the lymphocyte side, namely TIGIT, DNAM-1 and CD96[9], which also bind to CD112, but it is the interaction between TIGIT and CD155 that has been shown to contribute the most substantial inhibitory signaling on NK cells[10]. This is partly owing to the expression of an immunoreceptor tyrosine-based inhibitory motif in the cytoplasmic portion of TIGIT, which stimulates immunosuppression upon binding to CD155 on GBM cells. Yet, TIGIT monotherapy has so far resulted in divergent NK cell responses. Because co-expression of TIGIT correlates to other inhibitory receptors on immune cells[11,12], its apparent role as a co-receptor has most often benefited from dual blockade approaches. On NK cells, this has translated into the requirement of co-activation via IL-15 stimulation[13,14] or agonism of 4-1BB. We recently characterized TIGIT as a heterogeneous receptor on NK cells in GBM, fueled by evidence of its role in NK cell maturation and sensitivity to inhibitory interactions driven by CD155. In turn, this led to our hypothesis that effectiveness of TIGIT immunotherapy requires the shifting of TIGIT interactions away from immunosuppression involving its ligands, rather than the mere blockade of TIGIT.

We previously showed that local blockade of CD73 activity, via a cleavable antibody fragment, is an efficient way of not only suppressing inhibitory concentrations of adenosine in the GBM TME, but also promoting NK cell immunometabolic activity against this tumor[15,16]. Though CD155 and CD73[17] have been independently studied in gliomas, the dual-targeting of these ligands lacks functional characterization. In addition, GBM immunotherapy with adoptively transferred, engineered immune cells has been challenged by the dysfunctional phenotype and hypofunctional nature of these cells in GBM patients[18], necessitating better sources of effector-capable cells to maximize their potency and concentration in the tumor[19].

In this work, to combat such dysfunction, and to address the difficult task of sourcing and expanding GBM donor NK cells[20], we present a platform using induced pluripotent stem cell (iPSC)-derived NK (iNK) cells as allogeneic effectors as an example of responsively-engineered, dual-targeting iNK cells for GBM. To achieve selective "role reversal" of TIGIT and combine effective ablation of the CD155-TIGIT axis while promoting NK cell metabolic activity in GBM, we describe a synNotch[21]-based construct that responds to CD155 as a priming antigen to induce transcriptional activation that results in release of an antibody to block immunometabolic activity in GBM, in this case via adenosine-producing CD73. In this example, we achieve antigen recognition of CD155 by engineering iPSCs with synNotch genetic constructs, and subsequently differentiating these cells into functional NK cells. These engineered iPSC-NK cells induce significant anti-GBM responses in patient-derived orthotopic xenografts, and can be manufactured at scale while sustaining synNotch expression. Our genetically modified NK cells recognize and bind CD155 via a TIGIT extracellular ligand, the binding of which to CD155 not only blocks immunosuppressive TIGIT/CD155 interactions, but also promotes GAL4-VP64 transcription factor-mediated binding to an upstream activating sequence within the construct, to spur the translation and release of an anti-CD73 scFv directly within the GBM tumor microenvironment (TME). Such a construct acts as a "decoy" system by usurping TIGIT-CD155 interactions and in turn promoting NK cell killing of GBM. Usurping the TIGIT-CD155 axis via overexpression of a synNotch construct on NK cells not only takes over inhibitory signaling by redirecting TIGIT activity toward immune-promoting Notch-

activation, but it also precludes TIGIT from binding to its other binding partner, CD112, to induce secondary signaling on NK cells in the TME[22], as well as the involvement of CD155 in other inhibitory interactions otherwise possible with TIGIT blockade alone. It also avoids antigenic downregulation of CD155 on GBM cells as a means of "antigen escape" by masking TIGIT-ITIM as a stimulatory receptor. As a pre-clinical example of a therapeutic correlation between TIGIT and CD73 in GBM, our study demonstrates the benefit of these two pathways as therapeutic co-targets in GBM, and the superior potential of usurping their activities in contrast to pharmacological blockade alone. Our study paves the way for new clinical immunotherapy treatments, while expanding the current GBM-targeted therapies by characterizing a multi-functional, inducible engineered iNK cell therapy for treatment of GBM. With this we hope to encourage the development of more effective and innovative treatment options for GBM patients.

## Results
### CD155 and CD73 are highly expressed and correlated, and denote poor patient outcomes in GBM
Gene expression and Kaplan-Meier survival curves were generated for GBM based on TCGA data. These data show an upregulation of CD155 (*PVR*) and CD73 (*NT5E*) in GBM, compared with normal tissue and that co-expression of CD155 and CD73 acts as a negative prognostic factor in GBM (Fig. 1A, B, Supplementary Fig. 1). Further, using TCGA data, expression of CD155 and CD73 were correlated in GBM as well as normal brain cortex tissues, and were found to be moderately correlated in GBM, but uncorrelated in normal brain tissue, using Pearson's correlation (Fig. 1C). We confirmed upregulation of CD155 and CD73 in GBM tissue of human patients through immunofluorescent staining using a human tissue array containing cores from astrocytoma, GBM and normal healthy brain tissue (Fig. 1D–F). Though inherent heterogeneity in marker expression is a hallmark of GBM, we found that CD155 is significantly upregulated in GBM when compared to normal brain tissue and lower grade gliomas. We extracted and analyzed RNAseq data from the Cancer Genome Atlas (TCGA) and classified these data into high and low groups for CD155 (*PVR*) and CD73 (*NT5E*) genes. We show Venn diagrams depicting association of *PVR* and *NT5E* among high and low-expressing populations (Fig. 1G). These associations show co-expression of both *PVR* and *NT5E* in approximately half of the GBM patient samples analyzed, showing that co-expression of CD155 and CD73 is prominent in GBM and dual-targeting of both antigens may be a promising approach to targeted immunotherapy. Further, we show a differentially expressed gene set for *PVR/NT5E*$^{high/high}$ and *PVR/NT5E*$^{low/low}$ patient populations (Fig. 1H, Supplementary Data 1–2).

Expression of CD73 and CD155 was also measured on a variety of human GBM cell lines. GBM43 WT cell line expressed high levels of CD73 and CD155 (Fig. 1I, Supplementary Figs. 2–3). GBM43 CD155KO, GBM43 CD73KO, and GBM43 CD155/CD73KO cell lines were and confirmed to have >99% knockout purity of CD155 and/or CD73 following CRISPR gene editing, respectively (Supplementary Fig. 3). Further, proliferation capacity of KO GBM lines was assessed and compared with GBM43 WT cells in vitro. Knockout of CD73, with or without CD155 knockout, was shown to enhance proliferation of GBM43 cells, but knockout of CD155 did not significantly impact GBM43 proliferation in vitro (Supplementary Fig. 4).

### Engineered iNK cells are phenotypically similar to non-engineered cells throughout differentiation and expansion
Human iPS cell line hiPSC01 was engineered to express an inducible genetic construct based on the synNotch signaling system, designed to co-target CD155, via TIGIT, and CD73 (Fig. 2A, B). Briefly, this construct consists of an extracellular TIGIT ligand coupled with a synthetic Notch (synNotch) transmembrane domain, followed by a GAL4-VP64 transcription factor. Binding of TIGIT with CD155 induces a conformational

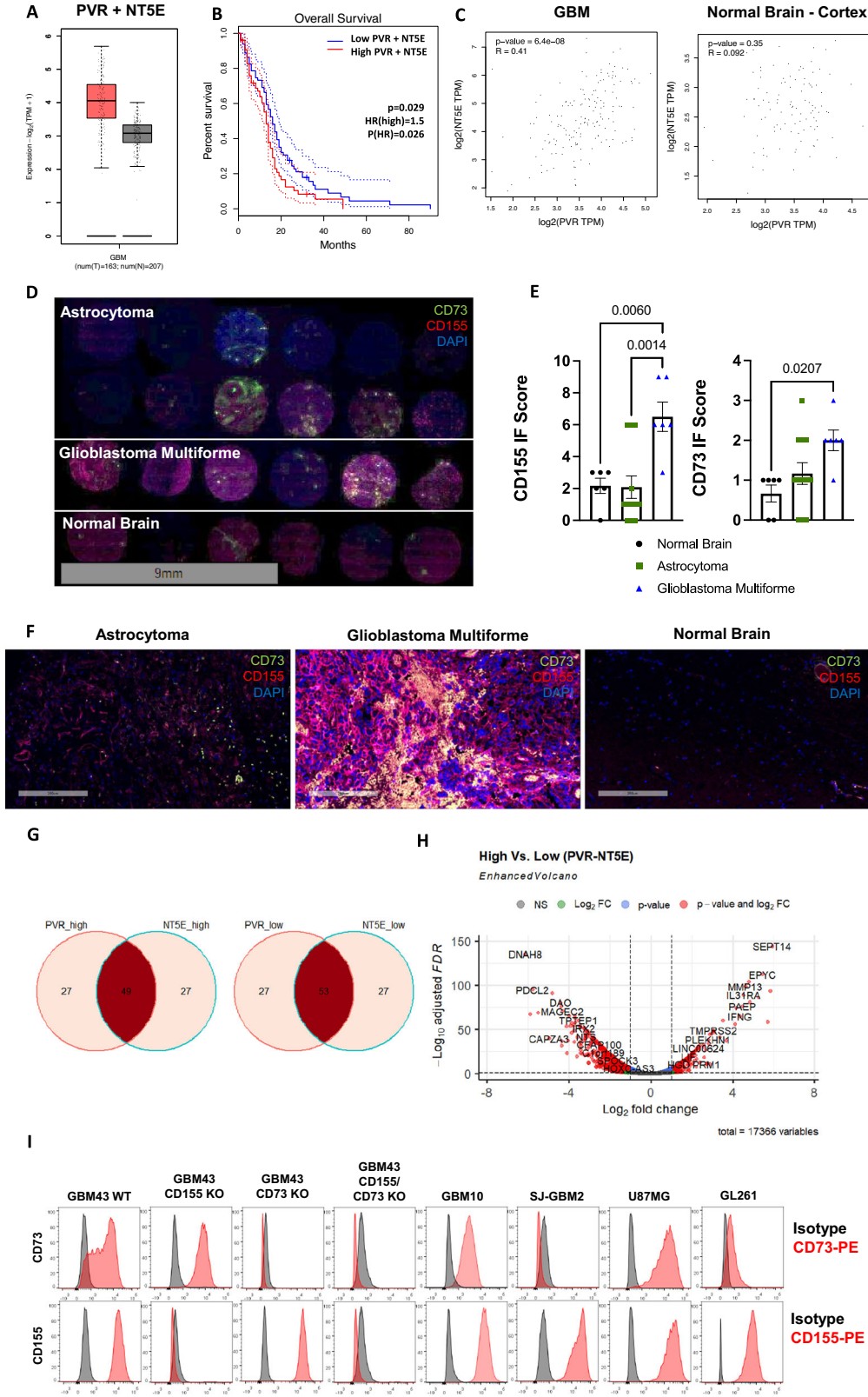

change within the synNotch domain, revealing a cleavable peptide, which is cleaved by tumor necrosis factor-alpha converting enzyme, or ADAM17 (Fig. 2B). Other proteases may also be responsible for this cleavage, but so far, this mechanism is the currently accepted model. This cleavage releases the GAL4-VP64 transcription factor, which in turn binds to an upstream activating sequence (UAS) on a second genetic construct, initiating the transcription, translation, and release

of an anti-CD73 scFv at the GBM-NK cell interface. Fully differentiated iNK cells are able to efficiently express the TIGIT.SynNotch.aCD73 genetic construct following lentiviral transduction, as measured by flow cytometry (Fig. 2C). WT and TIGIT.SynNotch.aCD73 iNK cells express TIGIT at similar levels, allowing us to accurately assess the blocking effect of the TIGIT.SynNotch construct against CD155 + GBM targets (Fig. 2D). Further, to ensure proper expression of our

**Fig. 1 | CD155 and CD73 are highly upregulated in GBM and represent negative prognostic factors. A** Transcriptional gene expression TCGA data depicting co-expression levels of *PVR* and *NTSE* in GBM versus normal tissue; Tumor−$n = 163$ patients (min: 1.9, max: 5.7, median: 4.1, lower whisker bound: 2.1, upper whisker bound: 5.7, Q1: 3.6, Q3: 4.6), Normal−$n = 207$ patients (min: 1.1, max: 4.0, median: 3.1, lower whisker bound: 2.1, upper whisker bound: 4.0, Q1: 2.8, Q3: 3.4). **B** Kaplan-Meier survival plot of *PVR/NTSE*$^{high/high}$ versus *PVR/NTSE*$^{low/low}$ GBM patients ($n = 81$ patients/group; Mantel-Cox test). **C** Two-tailed Pearson correlation of *PVR* and *NTSE* expression in GBM and normal brain tissue; Tumor−$n = 163$ patients, Normal−$n = 115$ patients. **D** IF tissue array staining for CD155 and CD73 expression in astrocytoma ($n = 12$ cores), GBM ($n = 6$ cores), and normal brain tissue ($n = 6$ cores). **E** IF scoring of staining in Fig. 1D (ordinary one-way ANOVA, Tukey's multiple comparison test). **F** Representative IF tissue array staining of CD155 and CD73 expression from Fig. 1D at 200x magnification. **G** Venn diagram of no. of patients in *PVR/NTSE*$^{high/high}$ and *PVR/NTSE*$^{low/low}$ groups from TCGA-GBM patient transcriptional gene expression data. ($n = 156$ patients). **H** Volcano plot of differentially expressed genes in *PVR/NTSE*$^{high/high}$ and *PVR/NTSE*$^{low/low}$ groups. **I** Histograms depicting CD73 and CD155 expression on GBM cell lines measured via flow cytometry. Data are presented as mean values +/- SEM. Source data are provided as a Source Data file.

TIGIT.SynNotch.aCD73 construct, primary peripheral blood NK cells were transduced to express our TIGIT.SynNotch.aCD73 construct and were stained intracellularly for the GAL4 transcription factor. Engineered NK cells expressed both mCherry and GAL4, while WT NK cells did not, demonstrating proper translation and expression of the TIGIT.SynNotch.aCD73 construct (Supplementary Fig. 5).

Further, following hematopoiesis, as well as following NK cell maturation and expansion from CD34$^+$ progenitors, iNK cells were assessed for phenotypic markers characteristic of hematopoietic progenitor cells and mature NK cells, respectively. Engineered iNK cells are able to initiate hematopoiesis and maintain expression of markers characteristics of hematopoietic and progenitor cells prior to lineage commitment (Fig. 2E, F, Supplementary Fig. 6). Following NK cell differentiation, engineered iNK cells can be generated at similar purity to WT iNK cells (Fig. 2G). Additionally, after 2 weeks of NK cell expansion, both engineered and WT iNK cells reach high levels of NK cell purity (>80%) (Fig. 2H). Variability amongst samples may be due to inherent process-related variability associated with generation NK cells from iPSCs.

Engineered iNK cells also express typical NK cell activating and inhibitory markers, at levels similar to those of non-engineered iNK cells both following differentiation and after expansion (Fig. 2I and K, Supplementary Fig. 7). Further, engineered iNK cells are able to expand at rates similar to those of non-engineered iNK cells, maintaining high viability throughout expansion (Fig. 2J and L).

### iNK cells, engineered with an inducible genetic construct can target CD155 and CD73-expressing patient-derived GBM cells

We first assessed the functional efficacy of dual blockade of CD73 and CD155/TIGIT using blocking antibodies. In the presence of both TIGIT and CD73 blocking antibodies, WT iNK cells were able to lyse GBM43 WT cells at significantly higher levels than without antibody blockade at high E:T ratios, suggesting there may be some dose-dependence on CD73 and TIGIT dual-targeting therapies. However, either CD73 or TIGIT blockade alone did not significantly enhance the cytolytic capacity of WT iNK cells against GBM43 target cells at high E:T ratios (Fig. 3A). Activation of the TIGIT.SynNotch.aCD73 construct was assessed against GBM43 WT (CD155 + CD73 + ) target cells and GBM43 CD155/CD73 KO target cells. An increase in GFP expression was measured on mCherry+ TIGIT.SynNotch.aCD73 iNK cells cocultured overnight with CD155 + CD73 + GBM43 WT target cells, but not GBM43 CD155/CD73 KO cells, when compared to unstimulated controls, indicating the activation of the aCD73 portion of our construct (Fig. 3B). Engineered iNK cells were then challenged in vitro with GBM cell lines expressing or lacking CD155 (generated via CRISPR/Cas9 knockout) and/or CD73 in cytolysis assays and with GBM43 WT cells for degranulation and cytokine production. TIGIT.SynNotch.aCD73-engineered iNK cells lysed GBM43 WT cells, which express both CD73 and CD155, at significantly higher levels than WT iNK cells (Fig. 3C). However, engineered iNK cells were equivalent to WT iNK cells in lysing GBM cells lacking these receptors, namely CD155KO, CD73KO, and double CD155/CD73KO cell lines, demonstrating the specificity of engineered iNK cells for CD73 and CD155 target ligands (Fig. 3D-F). Additionally,

TIGIT.SynNotch.aCD73 iNK cells demonstrated superior lysis against CD155 + CD73 + GBM10 and U87-MG targets, but not CD155 + CD73-SJ-GBM targets (Fig. 3G-I). These results demonstrated the requirement for the presence of both CD155 and CD73 engagement via the synNotch cascade for optimal activation of iNK cells, and suggesting minimal leakiness of the synNotch construct owing to the inability of engineered iNK cells to lyse targets without proper receptor/ligand activation. TIGIT.SynNotch.aCD73 iNK cells also demonstrated minimal cytolysis of normal cortical neuron cell line, HCN-2, over WT iNK cells, showing minimal on-tumor off-target toxicity (Supplementary Fig. 8). Additionally, TIGIT.SynNotch.aCD73 engineered primary peripheral blood NK cells were directly compared against WT NK cells with CD73 mAb and TIGIT mAb blockade in cytolysis of CD155 + CD73 + GBM43 target cells. Engineered NK cells demonstrated superior cytolysis of GBM43 target cells, demonstrating the enhanced therapeutic efficacy of our engineered NK cells (Supplementary Fig. 9).

Further, engineered iNK cells challenged with GBM43 WT cells expressed more CD107a and IFN-γ, and secreted more TNF-α than WT iNK cells, further demonstrating the functional enhancement of genetically engineered iNK cells (Fig. 3J-M). Differences in unstimulated controls may be explained by inherent variability associated with in vitro NK cell differentiation.

### Dual targeting of CD73 and CD155 dramatically slows tumor growth in an immunocompetent mouse model

Immunocompetent C57BL/6 mice were treated with or without CD155 and/or CD73 blocking antibodies for 3 weeks following implantation of GL261, mouse glioma, intracranial tumors (Fig. 4A). Dual treatment with both CD155 and CD73 blocking antibodies significantly slowed tumor growth compared to either antibody alone, or a PBS control group, demonstrating the efficacy of combined CD155 and CD73 targeting (Fig. 4B, C). Further, dual treatment with both CD155 and CD73 blocking antibodies had increased intracranial tumor infiltrates of NKp46+GzB+ NK cells and CD3 + T cells, without significant upregulation of PD-1 + T cells, LAG-3 or FoxP3+ Tregs, further justifying targeting of both tumor antigens (Fig. 4D–G, Supplementary Figs. 10–11).

In a separate study, immunocompetent C57BL/6 mice were treated with or without CD155 and/or CD73 blocking antibodies for 2.5 weeks following engraftment of GL261, mouse glioma, flank tumors (Supplementary Fig. 12A). Dual treatment with both CD155 and CD73 blocking antibodies significantly slowed tumor growth compared to either antibody alone, or a PBS control group, demonstrating the efficacy of combined CD155 and CD73 targeting (Supplementary Fig. 12B, C).

After treatment, immune cells were isolated from the blood, spleen, and tumor tissues of all mice and profiled via flow cytometry (Fig. 4H, I, Supplementary Figs. 13–15). Immune populations were first distinguished into monocyte, lymphocyte, and granulocyte populations based on CD45 staining, and subsequently separated into the following sub-populations: NK cells, (NK1.1 + CD3-), T cells (NK1.1-CD3 + ), CD4 + T cells, CD8 + T cells, regulatory T cells (CD4 + CD25+FoxP3 + ), neutrophils (F4/80 + Ly6G + ), M1 macrophages (F4/

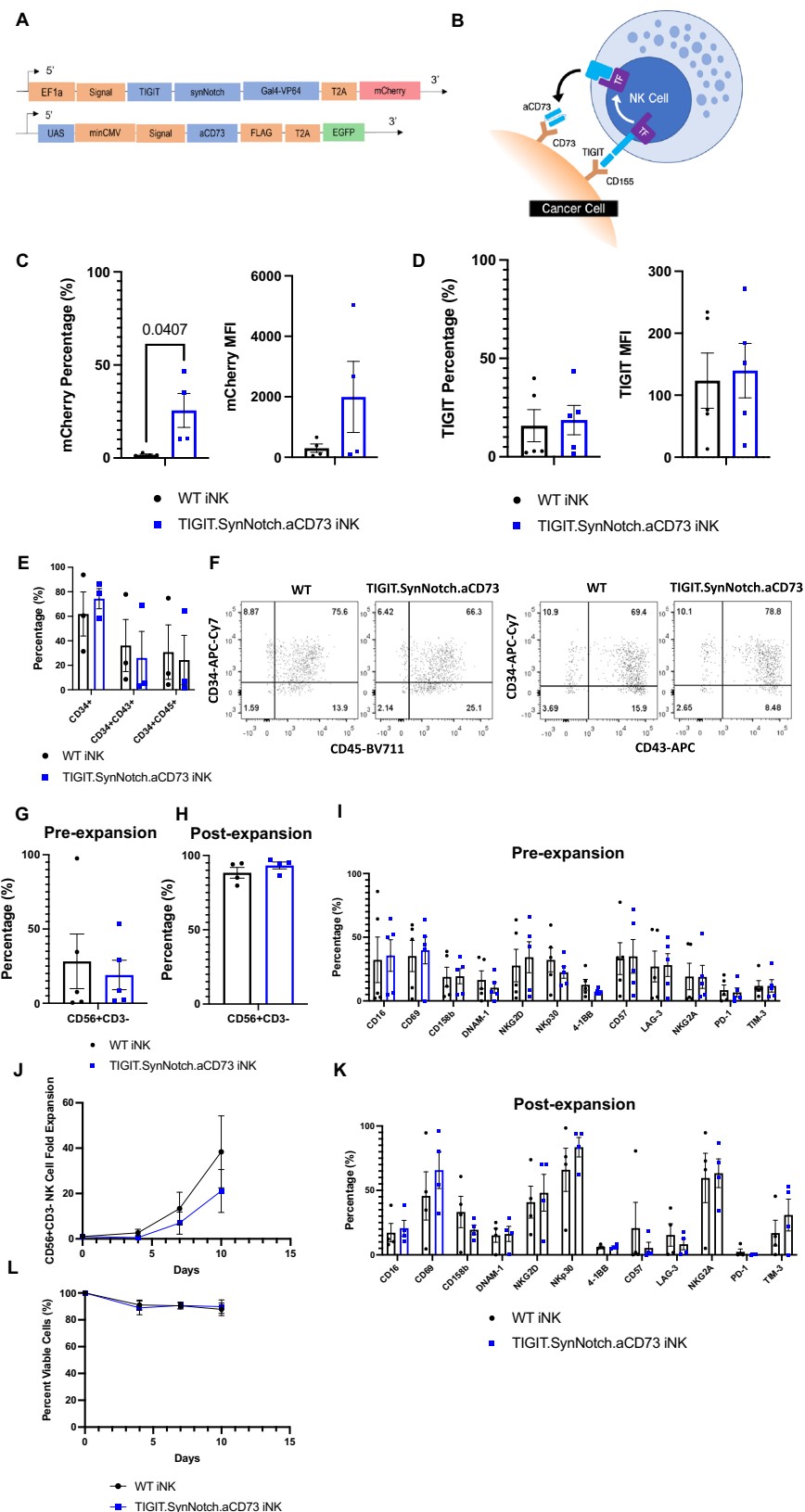

80 + MHC II+iNOS + ), M2 macrophages (F4/80 + CD206+Arg-1 + ). NK and T cells were further phenotyped for activation and exhaustion markers. Notably, there was an upregulation of tumor-infiltrating NK cells and CD8 + T cells in mice treated with both CD73 and CD155 mAbs compared to the PBS control, and a downregulation of CD4 + T cells (Fig. 4H). Mice treated with CD73 mAb alone were shown to have an increase in tumor infiltrating NK cells, but a decrease in CD8 + T cells. Further, we show an increase in anti-tumor M1 macrophages and no significant change in the pro-tumor M2 macrophage in the tumors of CD73/CD155 dual-treated mice. NK and T cell phenotypes were further analyzed for expression of various inhibitory and activating markers (Fig. 4I). We observed an upregulation of PD-1 in both NK and T cells

**Fig. 2 | iNK cells generated via a feeder-free differentiation protocol can be engineered to stably express a synNotch-based genetic construct. A** Diagram of TIGIT.SynNotch.aCD73 genetic construct structure consisting of TIGIT-SynNotch-GAL4-VP64 recognition domain and UAS-aCD73 inducible scFv. **B** Schematic diagram of TIGIT.SynNotch.aCD73 activation and subsequent secretion of aCD73 scFv directly at the NK:cancer interface. **C** Gene expression percentage (left) and MFI (right) of TIGIT.SynNotch.aCD73 on iNK cells following differentiation measured via mCherry expression via flow cytometry ($n = 4$ independent experiments; unpaired two-tailed $t$-test). **D** TIGIT expression percentage (left) and MFI (right) measured on TIGIT.SynNotch.aCD73-engineered iNK cells via flow cytometry ($n = 5$ independent experiments; unpaired two-tailed $t$-test). **E** Purity of CD34 + , CD34 + CD43 + , and CD34 + CD45+ cells following 11 days of hematopoietic progenitor differentiation ($n = 3$ independent experiments; multiple unpaired two-tailed $t$-

tests). **F** Representative dot plots of data depicted in Fig. 2E. **G** Engineered iNK cell purity following 39 days of NK cell differentiation ($n = 5$ independent experiments; unpaired two-tailed $t$-test). **H** Engineered iNK cell purity following 39 days of NK cell differentiation and an additional 2 weeks of iNK cell expansion ($n = 4$ independent experiments; unpaired two-tailed $t$-test). **I–K** Flow cytometry phenotype of engineered iNK cells following 39 days of NK cell differentiation ($n = 5$ independent experiments; multiple unpaired two-tailed $t$-tests) as well as following 2 additional weeks of iNK cell expansion ($n = 4$ independent experiments; multiple unpaired two-tailed $t$-tests). **J–L** Fold expansion and viability of engineered iNK cells for 10 days following NK cell differentiation ($n = 3$ independent experiments; two-way ANOVA, Sidak multiple comparison test). Data are presented as mean values +/- SEM. Source data are provided as a Source Data file.

---

from mice treated with CD73 mAb alone, and no significant upregulation of this receptor in the dual CD73/CD155 mAb treatment group. Although the phenotype of tumor-infiltrating NK cells upon dual CD73/CD155 treatment was comparable to that of NK cells treated with CD155 alone, dual-treated tumors showed a significantly larger proportion of NK cells infiltrating the tumors and a more robust anti-tumor response, indicating that phenotype alone does not define NK cell dysfunction.

### SynNotch-engineered iNK cells are highly active against GBM in an orthotopic, patient-derived xenograft model in immunodeficient mice

To assess the functional capacity of synNotch-CD73/CD155 co-targeted iNK cells against an intracranial model of GBM, we established a GBM43 xenograft orthotopic model in NSG mice. To do so, we genetically engineered GBM43 cells to express a firefly luciferase gene to enable active monitoring of tumor progression throughout the study[15]. NSG mice were first implanted intracranially with $1 \times 10^5$ GBM43-luciferase cells, then subsequently received 3 weekly injections of $2 \times 10^6$ WT or TIGIT.SynNotch.aCD73 engineered iNK cells, also intracranially (Fig. 5A). Tumor size and bodyweight were monitored throughout the study (Fig. 5B–E). Mice receiving engineered iNK cells recorded a drastic reduction in tumor growth, when compared with the PBS control group or mice receiving WT iNK cells, with many tumors nearly being eliminated altogether (Fig. 5B–E). Further, treatment with engineered iNK cells markedly improved survival over non-engineered iNK and control mice, demonstrating the potency of the CD73/CD155 co-targeting engineered NK cell therapy in treating GBM (Fig. 5F). Whole brain samples collected from PBS, WT iNK and TIGIT.SynNotch.aCD73 iNK treated mice were harvested and immunohistochemically stained for NKp46, granzyme B (GzB), CD73 and CD155. Mice treated with TIGIT.SynNotch.aCD73 iNK cells had higher numbers of tumor-infiltrating NKp46+ and GzB+ NK cells, demonstrating not only a higher infiltration of engineered NK cells into GBM brains, but a higher functional activation compared to WT iNK cells (Fig. 6A–D). Further, IHC samples were scored for CD73 and CD155 expression, and TIGIT.SynNotch.aCD73 iNK-treated mice exhibited lower levels of CD155 and CD73 than PBS or WT iNK-treated mice, demonstrating the ability of this genetic targeting to function in vivo and effectively downregulate both CD155 and CD73 within the tumor microenvironment (Fig. 6E–H). Overall, these data show the therapeutic efficacy of a localized dual-blockade of CD155 and CD73 within a synNotch activation cascade directly at the tumor-NK cell interface, and offer a new potential therapeutic strategy for treating GBM patients. In a separate study, NK cell homing to the brain was assessed following i.v. administration, and no NK cells were detected in the brain of tumor regions of treated mice, demonstrating the requirement for i.c. administration for therapeutic efficacy (Supplementary Fig. 16).

### Co-targeting CD73 and CD155 with genetically-engineered iNK cells enhances anti-tumor response over targeting either ligand alone

Immunodeficient NCG mice were engrafted with GBM43 WT flank tumors, and, after tumors were established, mice were treated with or without iNK cells according to the following treatment groups: WT, aCD73, TIGIT.SynNotch, and TIGIT.SynNotch.aCD73 (Supplementary Fig. 17A). Treatment of GBM43 tumors with iNK cells from any group reduced tumor progression over the PBS control group. Further, dual-targeted TIGIT.SynNotch.aCD73 iNK cells demonstrated enhanced anti-tumor responses over either WT or engineered iNK cells lacking the aCD73 domain, demonstrating the value of simultaneous targeting of CD73 and TIGIT in an adoptive transfer setting and the need for the transcriptional activation and translation of CD73-blocking scFv following TIGIT-CD155 engagement (Supplementary Fig. 17B, C). Tumors harvested from TIGIT.SynNotch.aCD73 iNK cell-treated mice showed a significant reduction in tumor weight when compared with the other treatment groups (Fig. 6I). Further, there was an upregulation in the number of infiltrating NK cells in TIGIT.SynNotch.aCD73-iNK-treated mice compared to WT iNK-treated mice, indicating an enhanced activity of dual-engineered iNK cells (Fig. 6J). Tumor-infiltrating NK cells harvested from TIGIT.SynNotch.aCD73-treated mice exhibited a reduction in PD-1, TIM-3, and 4-1BB compared to WT control groups, suggesting that these cells may be more functional and less functionally exhausted than control NK cells (Fig. 6K, L). Though single targeting of TIGIT exhibited similar expression of PD-1, single targeting was not sufficient to elicit potent anti-tumor responses.

## Discussion

GBM is one of the most aggressive and challenging tumors to treat, and is characterized by a high degree of heterogeneity, with varied expression of surface receptors, genetic markers, and immunosuppressive metabolites, all contributing to tumor and disease progression[23,24]. While checkpoint inhibition has been successful in boosting immune responses against a variety of tumors, GBM has remained largely resistant to such therapies. In GBM, resident immune cells, particularly NK cells, exist in an impaired and often functionally exhausted state from severe immunosuppression within the TME[25]. TIGIT, a receptor expressed on activated NK cells[8], contributes to such immunosuppression primarily through interactions with CD155[8]. Yet, TIGIT blockade monotherapy has yielded disappointing NK cell responses in solid tumors[26]. Part of this stems from, on one hand, the broad expression of TIGIT on activated and functionally mature NK cells and, on the other hand, the complex receptor-ligand network that TIGIT is involved in, necessitating impairing its inhibitory activities rather than merely blocking its binding capacity[27,28]. Within this context, we have developed an approach aimed at curbing TIGIT immunosuppression, via a genetic construct designed to usurp TIGIT activity and its interactions with CD155 on NK cells. The construct is based on the synNotch signaling system[29,30], which triggers

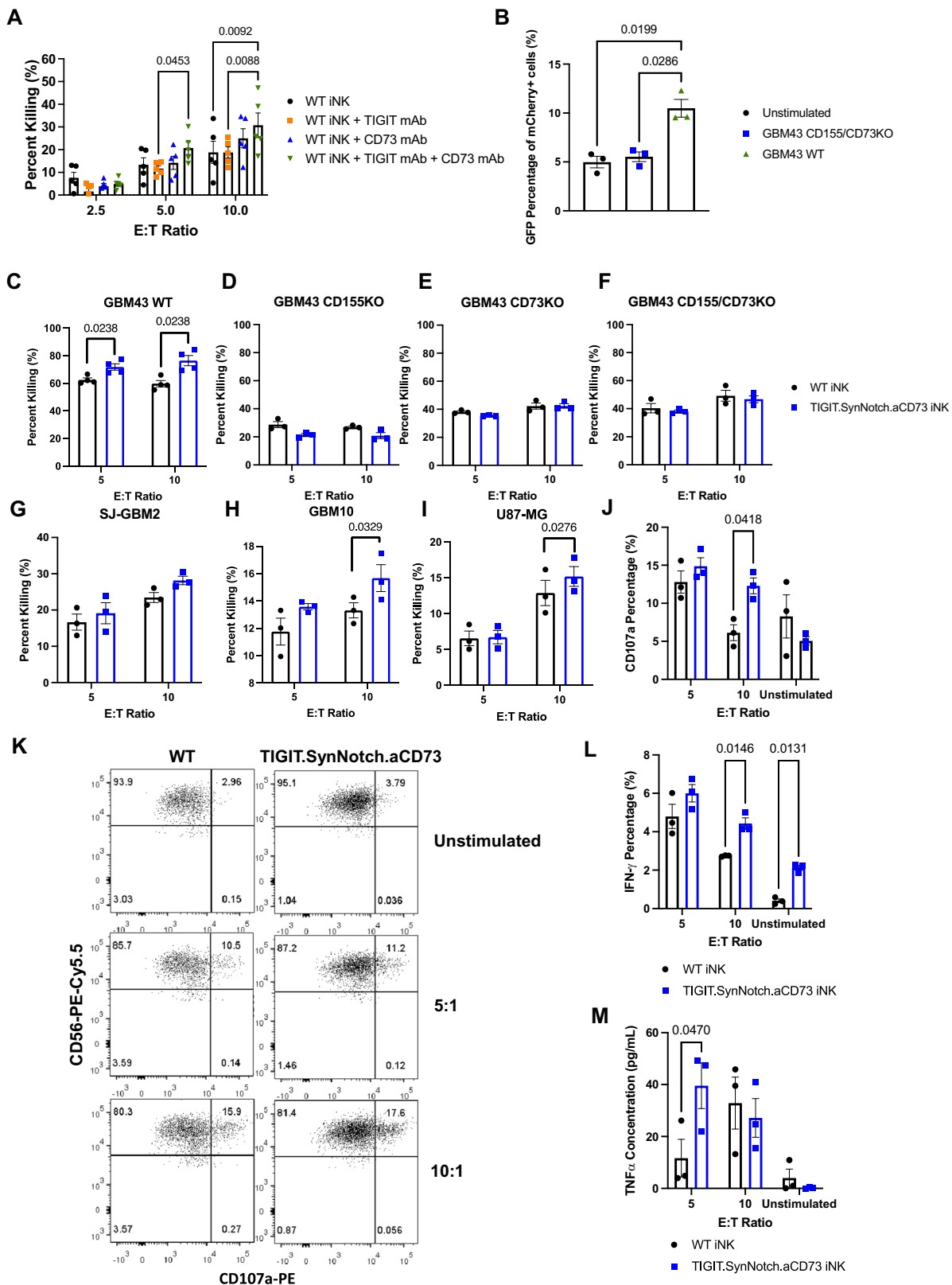

transcriptional activation of downstream translation machinery to release blocking scFv in the GBM TME. The net result is the shifting of TIGIT-CD155 immunosuppression toward NK cell activation without directly blocking its binding capacity. Such a "decoy" receptor engages in CD155 binding but usurps the natural signaling progression of this interaction, instead yielding GBM cell killing. This overcomes a major

limitation of TIGIT immunotherapy – the ability of CD155 to engage in alternative inhibitory functions upon blockade of TIGIT[31].

To generate functional NK cells deployable to immune-compromised GBM patients, a mechanism to overcome the dysfunction associated with immune cells harvested from these patients, which are characterized by impaired or abnormal phenotypes and

**Fig. 3 | Dual targeting of CD73 and CD155 enhances iNK cell activity against CD73 + /CD155 + GBM in vitro. A** Cytotoxicity of iNK cells with or without mAb blockade of TIGIT and/or CD73 against GBM43-WT primary patient derived GBM target cells ($n = 5$ independent experiments; two-way ANOVA, Tukey's multiple comparison test). **B** Activation of TIGIT.SynNotch.aCD73 genetic construct following overnight coculture with GBM43-WT or GBM43 CD155/CD73 KO cells detected via GFP expression via flow cytometry ($n = 3$ independent experiments; ordinary one-way ANOVA, Tukey's multiple comparison test). **C–I** Cytotoxicity of engineered iNK cells versus WT iNK against GBM targets: GBM43 WT ($n = 4$ independent experiments; multiple unpaired two-tailed $t$-tests), GBM43 CD155 KO, GBM43 CD73 KO, GBM43 CD155/CD73 KO, SJ-GBM2, GBM10, and U87-MG ($n = 3$ independent experiments; multiple unpaired two-tailed $t$-tests). **J, K** CD107a expression of iNK cells following 4 h coculture with GBM43 WT target cells ($n = 3$ independent experiments; multiple unpaired two-tailed $t$-tests). **L** IFN-γ expression by engineered iNK cells following 4 h of coculture with GBM43 WT target cells ($n = 3$ independent experiments; multiples unpaired two-tailed $t$-tests). **M** TNF-α secretion by engineered iNK cells following 24 h of coculture with GBM43 WT target cells ($n = 3$ independent experiments; multiple unpaired two-tailed $t$-tests). Data are presented as mean values +/- SEM. Source data are provided as a Source Data file.

functional and metabolic deficiency, thus limiting efficacy of autologous cell therapies[32–36], is needed. Metabolic deficiency is largely induced by immunosuppressive metabolites present within the GBM TME, such as adenosine, while functional impairment includes phenotypic abnormality of NK cells and their deficiency in performing effector functions, including target killing and cytokine production. This presents a need for an allogeneic platform of adoptively transferred NK cells that can target and combat immunosuppression within the GBM tumor microenvironment while avoiding the reliance on hypofunctional NK cells from the patient.

Thus, in the current study, we deployed iPSC-derived NK cells[37,38], a potent cellular platform with the potential for allogenicity. Using these cells, generated under chemically-defined, feeder-free conditions and readily amenable to xeno-free manufacturing processes[37], we describe a GBM immunotherapy based on genetically engineered, allogeneic iPSC-derived NK cells. By using iPSC-derived NK cells, we have addressed two key challenges associated with cell-based immunotherapy for GBM: poor ex vivo expansion and genetic manipulation of primary NK cells, and limitations associated with functional impairment of autologous NK cells. SynNotch-based activation of NK cells against GBM via TIGIT achieves the local blockade of CD155 and CD73, directly within the GBM TME, thereby mitigating any off-tumor effects and antigen escape, and maximizing the efficacy of targeting GBM-associated checkpoints, by delivering locally-blocking ligands and antibody fragments directly to the GBM-NK cell interface. Though activation of the SynNotch construct, measured by GFP expression, yielded a modest response in vitro, in vivo responses were robust and durable. This may be due to the relatively low GBM:NK cell ratio in vitro compared to those present within the tumor microenvironment or in vitro lysis of GBM43 cells resulting in a reduced antigen exposure over time. Nonetheless, our robust in vivo responses, even with relatively low synNotch-activated iNK cells, suggests that synNotch-activated NK, secreting CD73 blocking scFv, may relieve even non-synNotch-activated NK cells of immune-inhibitory mechanisms, a further justification of this therapeutic strategy. Further, though there is potential for antigen escape mechanisms within GBM, our TIGIT.SynNotch.aCD73 iNK cells do not rely on target expression for tumor lysis, but instead on the natural toxicity of our iNK cells, limiting this immune-antigen escape.

We selected CD155 and CD73 as targets, due to their overexpression in GBM, and negative roles in patient prognosis[39–41]. Additionally, while many have explored targeting these ligands separately, the dual-targeting of CD155 and CD73 with an engineered cell therapy had not been explored. Co-targeting CD155 and CD73 relieves the GBM microenvironment from two major sources of immune resistance: a potent immunosuppressive axis which directly inhibits NK cell function, and metabolic reprogramming of the local TME that directly impacts metabolic fitness. In addition, in vivo inhibition studies showed that co-targeting these two receptors results in favorable reprogramming of the GBM TME, via enhanced recruitment of CD8+ T cells and concomitant loss of M2 macrophages. This was associated with phenotypic changes on effector immune cells, including a marked downregulation of PD-1 on NK cells. Conversely, targeting either alone has limited therapeutic efficacy. Here we show, through bioinformatics analyses of transcriptional patient data, in vivo efficacy and survival analysis, as well as pharmacological blockade assays, that targeting of CD155/TIGIT and CD73 carries substantial anti-GBM potential, with dramatic preclinical responses. In our experience, synNotch-based engineering of iPSC-NK cells was robust with minimal leakiness[30], and strongly relied on the presence of targeted antigens on cancer cells. The unique aspect of the iPSC-NK platform is the robustness and relative ease of engineering iPSCs[42–44] in the undifferentiated state—compared with the routine difficulty at engineering human blood-derived NK cells—in addition with these cells' ability to retain transduced gene expression throughout hematopoiesis and differentiation into mature immune cells. These engineered iPSC-NK cells are phenotypically indistinguishable throughout all phases of differentiation and expansion yet elicit potent effector functions against CD155+CD73+ GBM cells both in vitro and in vivo. Though single targeting of either TIGIT was able to partially ameliorate upregulation of exhaustion markers, dual targeting of TIGIT and CD73 was necessary to elicit robust tumor control (Figs. 5–6, Supplementary Fig. 17). Further, though in vitro functional assays yielded only modest improvements in GBM killing, intracranial in vivo tumor growth was dramatically reduced, likely due to TME specific factors, not present in in vitro systems. Additional modifications to generate fully allogeneic and patient-compatible, hypoimmunogenic iPSCs[45] are possible through HLA editing[46,47].

This study unveiled a powerful pathway to exploiting TIGIT activity while overcoming the divergent responses induced by its mere blockade, and demonstrated that iNK cells engineered to release a localized multi-targeted checkpoint blockade can elicit effective anti-GBM activity, relying upon the natural cytotoxicity of the iPSC-NK cells, without any additional CAR signaling. We believe the findings of this study will unveil multi-targeted iPSC-NK cells as a platform for allogeneic adoptive transfer immunotherapies, and reveal targetable TIGIT-CD155 interactions as combination therapies, thus paving the way for meaningful clinical advancements in NK cell therapies for GBM.

## Methods

All experiments were conducted in compliance with all relevant ethical regulations and standards. All animal experiments described in this study, utilizing C57BL/6, NOD.Cg-Prkdc[scid] IL2rg[tm1Wjl]/SzJ (NSG), and NOD.Cg-Prkdc[em26Cd52] IL-2[em26Cd22]/NjuCrl (NCG) mice, were approved by the Purdue University Animal Care and Use Committee. Mice were housed in a 12 h/12 h light/dark cycle, with a temperature of 18−−23 °C and relative humidity of 40-60%.

### Mice

Male and female 6–8 week-old C57BL/6 (The Jackson Laboratory, Bar Harbor, ME), NOD.Cg-Prkdc[scid] IL2rg[tm1Wjl]/SzJ (NSG; The Jackson Laboratory, Bar Harbor, ME), and NOD.Cg-Prkdc[em26Cd52] IL-2[em26Cd22]/NjuCrl (NCG; Charles River Laboratories, Fairfield, NJ) mice were maintained by our lab or at the Purdue Center for Cancer Research. Primary human NK (PNK) cells were obtained from healthy adult donors approved under Purdue University's Institutional Review Board (IRB) (IRB-approved protocol #1804020540).

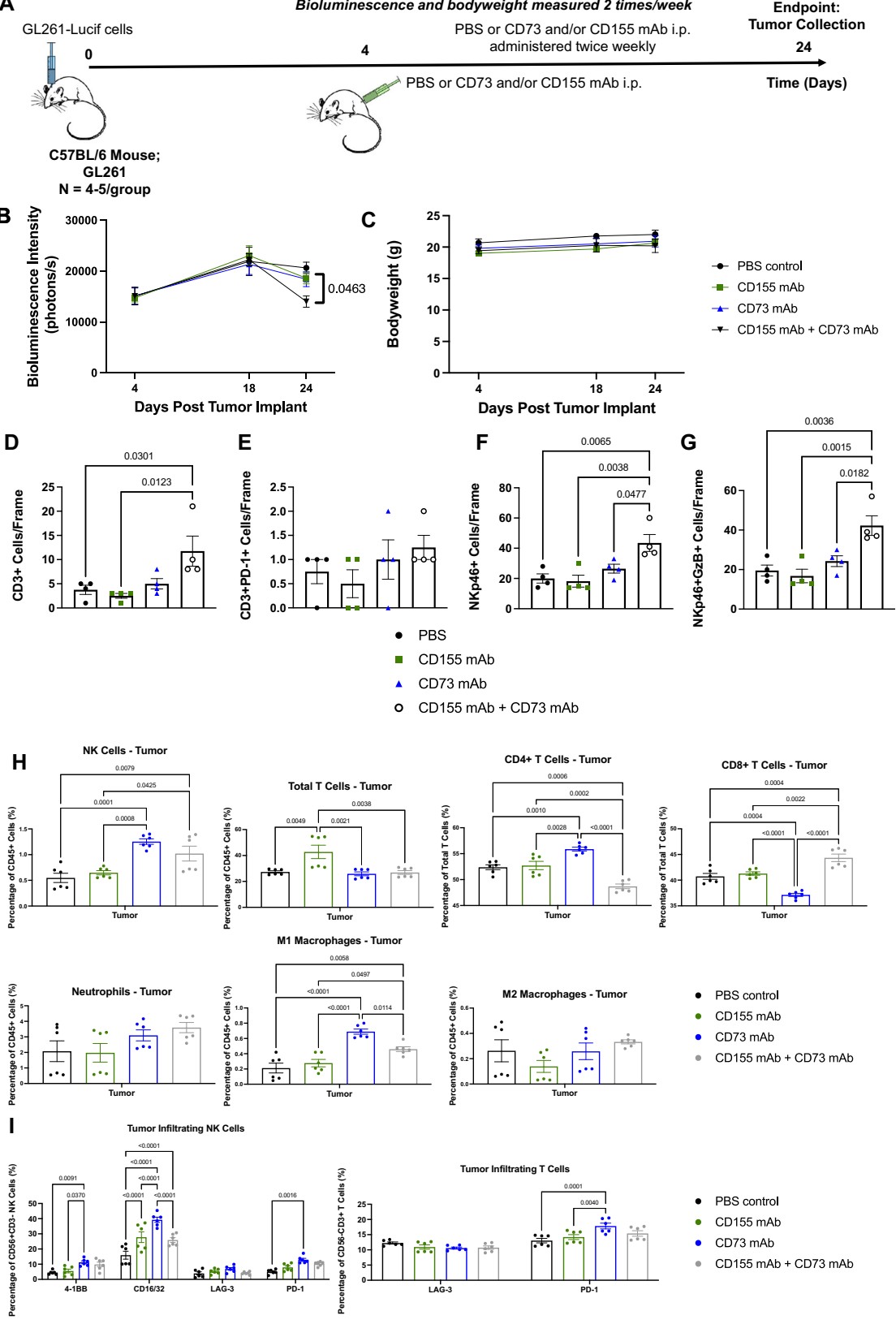

## Cell culture

GL261 (PTA-5893) and HCN-2 (CRL-3592) cells (purchased from ATCC) were maintained in DMEM (Gibco) supplemented with 10% FBS (Corning), 100 U/mL penicillin and 100 ug/mL streptomycin (Gibco) until passage 10 before being discarded. U87-MG (HTB-14) cells (purchased from ATCC) were maintained in IMDM (Gibco) supplemented

with 10% FBS, 100 U/mL penicillin and 100 ug/mL streptomycin until passage 20 before being discarded. SJ-GBM2 cells (kindly provided by Dr. Karen Pollok, Indiana University School of Medicine) were originally obtained from the Children's Oncology Group and were maintained in IMDM supplemented with 20% FBS, 1% ITS (Gibco), 100 U/mL penicillin and 100 ug/mL streptomycin until passage 10 before being

**Fig. 4 | Dual blockade of CD73 and CD155 enhances anti-tumor immune responses in immunocompetent GBM mouse model. A** Study timeline of GL261 intracranial tumor-bearing mice treated with or without CD73 and/or CD155 mAb. **B** Bioluminescence intensity and **C** bodyweight for intracranial GL261 tumor-bearing mice following treatment with PBS (*n* = 4 mice), CD155 mAb (*n* = 4 mice), CD73 mAb (*n* = 5 mice), or both CD155 and CD73 mAbs (*n* = 5 mice) (two-way ANOVA, Tukey's multiple comparison test). Quantification of tumor infiltrating **D** CD3 + T cells **E** CD3 + PD-1 + T cells for intracranial GL261 tumor-bearing mice following treatment with PBS, CD155 mAb, CD73 mAb or both CD155 and CD73 mAbs (*n* = 4 intratumoral sections; ordinary one-way ANOVA, Tukey's multiple comparison test) (Supplementary Fig. 10). Quantification of tumor infiltrating **F** NKp46+ NK cells **G** NKp46+GzB+ NK cells for intracranial GL261 tumor-bearing

mice following treatment with PBS, CD155 mAb, CD73 mAb or both CD155 and CD73 mAbs; GzB Granzyme **B** (*n* = 4 intratumoral sections; ordinary one-way ANOVA, Tukey's multiple comparison test) (Supplementary Fig. 10). **H** Percentage of various immune cells measured on tumor-infiltrating CD45+ cells in treated immunocompetent flank tumor-bearing mice measured via flow cytometry (*n* = 6 mice/group; ordinary one-way ANOVA, Tukey's multiple comparison test) (Supplementary Fig. 12–13). **I** Phenotypic analysis of tumor-infiltrating NK and T cells isolated from treated mice measured by flow cytometry (*n* = 6 mice/group; two-way ANOVA, Tukey's multiple comparison test, simple row effects) (Supplementary Figs. 12–13). Data are presented as mean values +/- SEM. Source data are provided as a Source Data file.

---

discarded[48,49]. GBM43 and GBM10 cells (kindly provided by Dr. Karen Pollok, Indiana University School of Medicine) DMEM without sodium pyruvate (Gibco) supplemented with 10% FBS, 100 U/mL penicillin and 100 ug/mL streptomycin until passage 10 before being discarded. GBM43 and GBM10 xenograft tissues were provided by Dr. Jann Sarkaria (Mayo Clinic, Rochester MN), and cell lines were subsequently derived from xenograft tissues in Dr. Pollok's lab. GL261, HCN-2, U87-MG, and K562 cell identity was confirmed by morphology. SJ-GBM2, GBM10, and GBM43 cell identity was confirmed by DNA fingerprint analysis (IDEXX BioResearch) for species and baseline short-tandem repeat analysis testing and were found to be entirely human[50]. All cells were maintained at 37 °C and 5% $CO_2$ in a humidified incubator. The hiPSC01 iPS cell line was generated by us at the Purdue Institute for Integrative Neuroscience Cell Engineering core facility. hiPSC01 cell identity was confirmed by STR analysis. Healthy male dermal fibroblasts (ReprocellUSA) were reprogrammed using the StemRNA 3rd Generation Reprogramming kit (Reprocell USA) according to the manufacturer's protocol. STR analysis for iPS cell lines has been shown for this line[37]. iPSCs were maintained and passaged in mTesR™ 1 (StemCell Technologies, 85850) feeder-free maintenance medium on hESC-Qualified Matrigel (Corning, 354277) coated 6-well plates. All cell lines are mycoplasma negative.

## Human NK cell isolation from peripheral blood and culture

Whole blood was collected in heparinized tubes (BD Vacutainer, 367874, BD, Franklin Lakes, NJ), untested for pathogens, and PNK cells were harvested from whole blood of adult healthy donors by negative selection, using the EasySep Direct Human NK cell Isolation Kit (StemCell Technologies, Vancouver, Canada). PNK cells were expanded for two to 4 weeks through coculture with K562 feeder cells (ATCC CLL-243, Manassas, VA), treated for 3 h prior to co-culture with mitomycin C (New England Biolabs, 51854 S, Ipswich, MA) to inhibit proliferation, at a 2:1 K562:NK cell ratio in RPMI media (Gibco, 21875034, Billings, MT) supplemented with 10% FBS (Corning, 35-011-CV, Corning, NY), 1% penicillin/streptomycin (Gibco, 15140122, Billings, MT), 50 ng/mL rh4-1BBL (Peprotech, 310-11, Cranbury, NJ), 50 ng/mL rhIL-21 (Gold Biotechnology, 1110-21, St. Louis, MO), and 500 IU/mL rhIL-2 (Akron Biotech, Boca Raton, FL). Prior to addition to PNK cells K562 cells were treated with 50 μg/mL of mitomycin C (Cayman Chemical, 11435, Ann Arbor, MI) for 3 h and wash with 1x PBS (Thermo Fisher Scientific, 70011044, Waltham, MA).

## Antibodies, stains, and reagents

Antibodies used for flow cytometry and cell sorting were CD3-PeCy7 (BD, 563423, clone UCHT1), CD56-PeCy5.5 (Thermo, 35-0567-42, clone CMSSB), TIGIT-APC (Biolegend, 372705, clone A15153G), DNAM-1-BV510 (Biolegend, 338329, clone 11A8), Sytox Green (Thermo, S34860), Sytox Blue (Thermo, S34857), CD16-BUV395 (BD, 563785, clone 3G8), NKG2A-FITC (Miltenyi, 130-114-091, clone REA110), PD-1-BV510 (Biolegend, 329931, clone EH12.2H7), NKG2D-BV605 (Biolegend, 320831, clone 1D11), CD57-BV605 (Biolegend,

393303, clone QA17A04), CD69-BV650 (Biolegend, 310933, clone FN50), LAG-3-BV650 (Biolegend, 369315, clone 11C3C65), NKp30-BV711 (Biolegend, 325217, clone P30-15), TIM-3-BV711 (Biolegend, 345023, clone F38-2E2), CD94-PE (Biolegend, 305506, clone DX22), CD158b-APC/Fire750 (Biolegend, 312617, clone DX27), 41BB-APC/Fire750 (Biolegend, 309833, clone 4B4-1), CD96-PE (Biolegend, 338405, clone NK92.39), CD56-APC (Thermo, 17-0567-42, clone CMSSB), CD107a-PE (Biolegend, 328607, clone H4A3), IFNy-PerCPCy5.5 (Biolegend, 502525, clone 4 S.B3), CD34-APC/Fire750 (Biolegend, 343535, clone 581), CD43-APC (Biolegend, 343205, clone CD43-10G7), CD45-BV711 (Biolegend, 304049, clone HI30), CD155-PE (Biolegend, 337609, clone SKII.4), CD73-PE (Biolegend, 344003, clone AD2), mCD155-PE (Biolegend, 132205, clone 4.24.1), mCD73-APC (Biolegend, 127209, clone TY/11.8), m4-1BB-APC (Biolegend, 106109, clone 17B5), mCD16/32-BV421 (Biolegend, 101331, clone 93), mNK1.1-BV510 (Biolegend, 108737, clone PK136), mCD3-BV711 (Biolegend, 100349, clone 145-2C11), mLAG-3-BV785 (Biolegend, 125219, clone C9B7W), mCD4-APC-Fire750 (Biolegend, 100459, clone GK1.5), mCD8a-PE-Dazzle594 (Biolegend, 100761, clone 53-6.7), mCD25-BV785 (Biolegend, 102051, clone PC61), mFoxP3-BV421 (Biolegend, 126419, clone MF-14), mPD-1-PE-Dazzle594 (Biolegend, 135227, clone 29 F.1A12), mF4/80-BV510 (Biolegend, 123135, clone BM8), mMHC II-BV605 (Biolegend, 107613, clone M5/114.15.2), miNOS-PE (Biolegend, 696805, clone W16030C), mCD206-BV650 (Biolegend, 141723, clone C068C2), mArginase-1-APC (Thermo Fisher, 17-3697-82, clone A1exF5), mLy6G-PE-Cy7 (Biolegend, 127617, clone 1A8), GAL4 (Thermo, 33-8600), and Goat anti-mouse APC (Biolegend, 405308, clone Poly4053). Transcription factor staining was done using the Nuclear Transcription Factor staining kit (Biolegend, 424401). Tumor samples were dissociated using DNAse I and Collagenase IV purchased from Worthington Biochemical. Samples were run on a BD Fortessa and analyzed using FlowJo V10. Prior to use antibodies were validated for staining of primary NK cells or murine immune cells, respectively, and all antibodies were used at a 1:200 dilution.

Antibodies used in functional assays and in vivo assays were mouse anti-PVR (Leinco, C2833, clone 4.24.1), mouse anti-CD73 (BioXCell, BE0209, clone TY/23), anti-TIGIT (Biolegend, 613704, clone A15153A), and anti-CD73 (Thermo, 41-0200, clone 7G2). Human rhIL-15 was from Shenandoah Biotechnology. Dilutions used are described in the appropriate experimental methods sections.

For histological analysis following intracranial xenograft studies, whole brain samples were fixed, paraffin embedded, sectioned and stained for NKp46 (Abcam, ab224703, clone EPR22403-57), NKp46 (Abcam, ab283505, clone Mncr1.05), granzyme B (Abcam, ab4059), CD73 (Cell Signaling Technology, 13160, clone D7F9A), CD73 (Biolegend, 344002, clone AD2), CD155 (Cell Signaling Technology, 13544, clone D3G7H), CD3 (Abcam, ab11089, clone CD3-12), PD-1 (Abcam, ab214421, clone EPR20665), FoxP3 (Abcam, ab215206, clone EPR22102-37), and LAG-3 (Abcam, ab237720, clone CAL77). Additionally, human brain tissue array

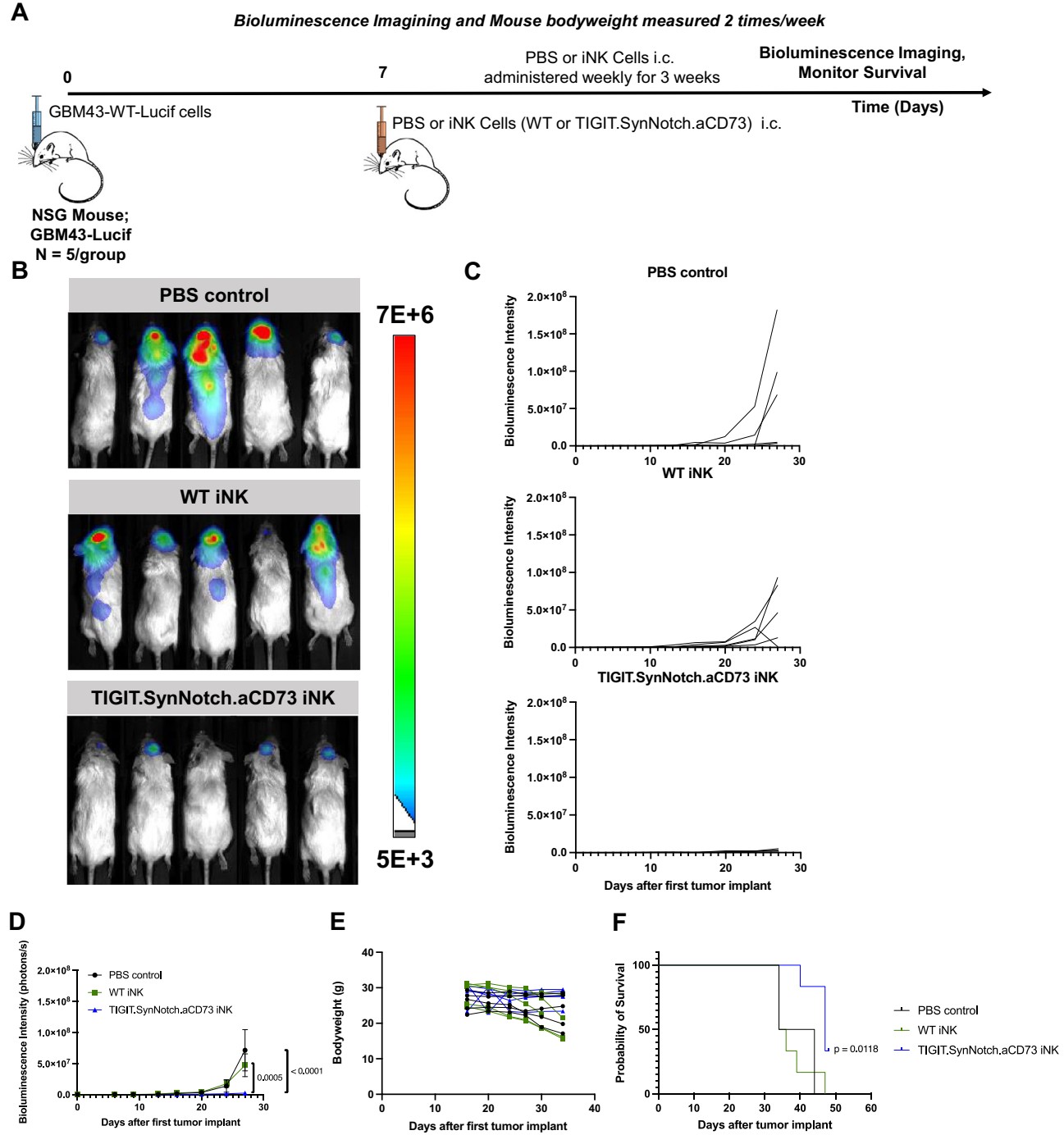

**Fig. 5 | Usurping CD155 and CD73 activities via synNotch activation of iNK cells can nearly halt tumor growth in an orthotopic patient-derived GBM mouse model. A** Study timeline of GBM43 tumor-bearing mice treated with or without engineered iNK cells. **B-C** Bioluminescence imaging of GBM43 WT tumor-bearing mice and measurement of bioluminescence intensity for individual mice throughout the course of the study ($n = 5$ mice/groups; two-way ANOVA, Tukey's multiple comparison test). **D-E** Average bioluminescence intensity and bodyweight measurements of mice measured throughout the course of the study ($n = 5$ mice/group). **F** Kaplan–Meier survival plot of PBS, WT iNK or synNotch-engineered iNK-treated tumor-bearing mice ($n = 6$ mice/group; Mantel-Cox test). Data are presented as mean values +/- SEM. Source data are provided as a Source Data file.

(GL242a) was purchased from US Biomax and stained for CD155 and CD73. Prior to use antibodies were validated for staining tissue, and all antibodies were used at a 1:200 dilution.

**Plasmid sequences and lentiviral production**
(1), and (2) were custom cloned and synthesized by VectorBuilder. (3) was from GenTarget Inc.
 (1) Plv[Exp]-Puro-EF1A>{TIGIT-Notch-Gal4(ns)}:T2A:mCherry

In this vector, the transgene sequence consisted of a signal peptide, a human TIGIT extracellular protein, coupled with a human synthetic notch protein (SynNotch) transmembrane domain and GAL4-VP64 transcription factor. This transgene was expressed under the control of an EF1alpha promoter, followed by a T2A sequence and mCherry fluorescent marker.
 (2) Plv[Exp]-Neo-5XUAS:miniCMV>Signal-CD73(human opt)/FLAG:T2A:EGFP

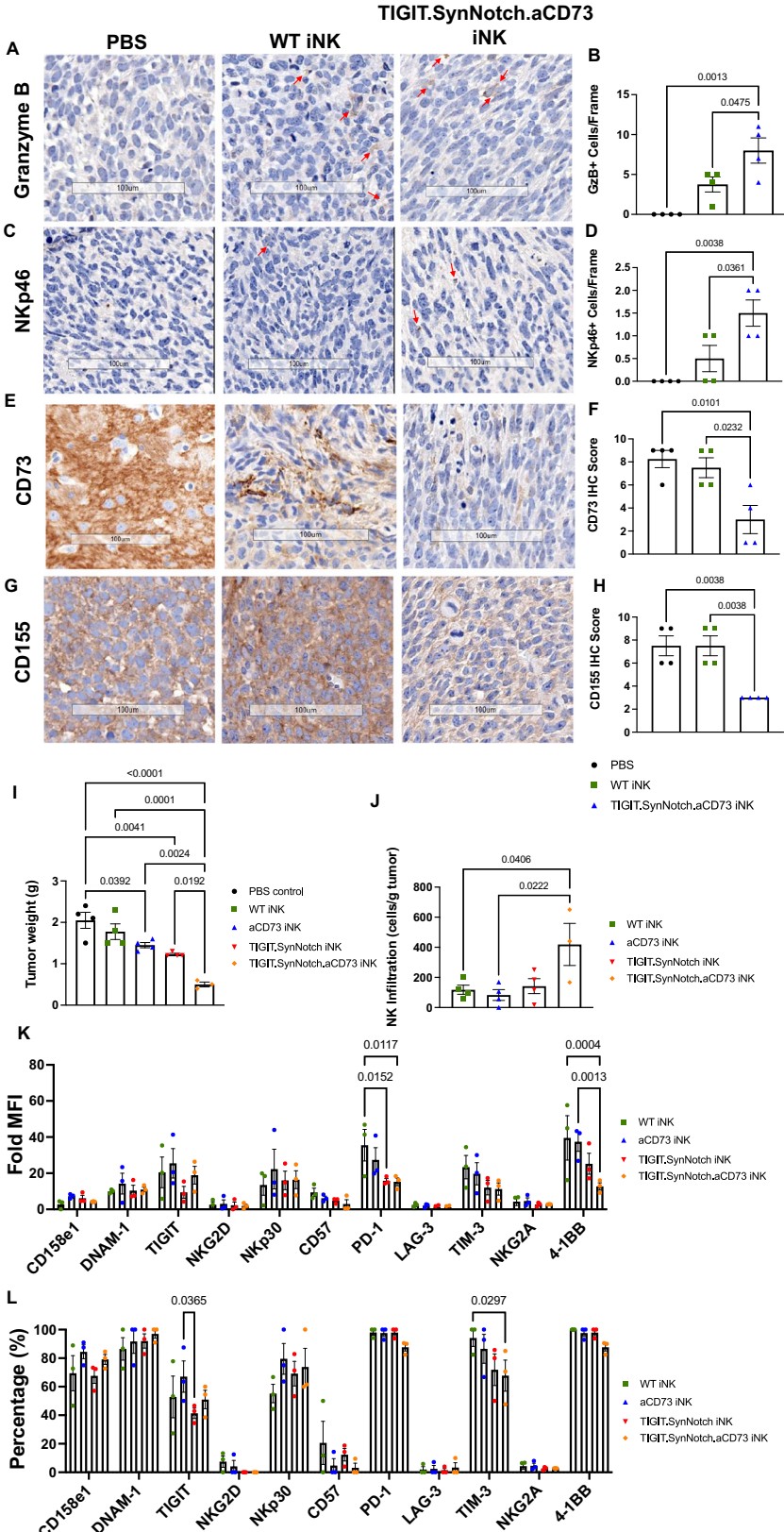

In this vector, the transgene sequence consisted of a signal peptide, and anti-CD73 scFv sequence, followed by a flag tag, T2A sequence, and EGFP fluorescent marker. The anti-CD73 scFv was based on previous work[15]. This transgene was expressed under the control of a minimal CMV promoter (miniCMV) and upstream activating sequence (UAS).

Vectors (1) and (2) were synthesized using a third-generation lentiviral packaging system. Proprietary envelope plasmid VSV-G, and packaging plasmids Gal/Pol and Rev were co-transfected along with each transgene into HEK293T (ATCC, CRL-3216) packaging cells using PEI. After 48 h, supernatant was collected, filtered, and lentiviral particles were concentrated with PEG. Following concentration and

**Fig. 6 | TIGIT.synNotch.aCD73-iNK cells demonstrate enhanced infiltration and reduced exhaustion over single-target controls. A–H** Representative IHC staining (200 x magnification) of GzB, NKp46, CD73, and CD155 in whole brain sections harvested from intracranial tumor-bearing mice treated with PBS, WT iNK or synNotch-engineered iNK cells, and quantification/scoring of IHC staining; GzB Granzyme B ($n = 4$ intratumoral sections; ordinary one-way ANOVA, Tukey's multiple comparison test). **I** Tumor weight of GBM43 WT flank tumors harvested from iNK or PBS-treated mice ($n = 4$ mice/group; ordinary one-way ANOVA, Tukey's

multiple comparison test) (Supplementary Fig. 17). **J** Quantification of NK cell infiltration in GBM43 WT flank tumors ($n = 4$ mice/group; ordinary one-way ANOVA, Tukey's multiple comparison test) (Supplementary Fig. 17). **K** Fold MFI and **L** percentage of NK receptor expression measured on iNK cells harvested from treated GBM43 WT flank tumors ($n = 3$ mice/group; two-way ANOVA, Tukey's multiple comparison test, simple row effects) (Supplementary Fig. 17). Data are presented as mean values +/- SEM. Source data are provided as a Source Data file.

resuspension in PBS, lentivirus was titered by p24 ELISA (Cayman Chemical) and stored at −80 °C.

(3) CMV-Luciferase (firefly), (Puro) Lentiviral Particles (LVP325)

Firefly luciferase expressing GBM43 cells were generated according to manufacturer recommendations.

### Generation of knockout GBM43 cell lines
Crispr/Cas9 gene knockout of CD73 and/or CD155 patient-derived GBM43 cells was carried out using the Gene Knockout Kit v2 (Synthego) according to manufacturer instructions. Briefly, sgRNA and Cas9 protein were mixed with Lipofectamine CRISPRMAX™ Cas9 Transfection Reagent (Thermo Fisher), incubated for 10 min at RT, and cells were transfected according to manufacturer instructions. Following gene knockout of CD73, cells were sorted using a BD Aria III for expression of >99% knockout purity, measured by CD73 cell surface expression. These cells were expanded, cryopreserved, and this knockout was repeated for CD155, using either GBM43 WT cells or GBM43 CD73KO cells, to generate GBM43 CD155KO and GBM43 CD155/CD73 double KO cell lines. All cell lines were cultured under the same conditions as GBM43 WT cells and used for functional assays.

### GBM cell line phenotyping and proliferation
GBM43 WT and knockout cell lines, along with other GBM cell lines (SJ-GBM2, U87MG, GBM10, and GL261) were stained for CD155 and CD73 and surface expression was measured via flow cytometry. For proliferation assays, GBM43 WT and knockout cell lines were seeded at 1000 cells/well in a standard cell culture 96-well plate. Proliferation was measured using the CCK-8 proliferation assay kit (Apex Bio) according to manufacturer instructions.

### TCGA analysis
Survival analysis based on expression of CD155 (PVR) and CD73 (NT5E) was done in GEPIA2 using TCGA gene expression data. Data were analyzed and generated using a Kaplan-Meier curve for overall survival (OS). Kaplan-Meier curves were generated with a median or quartile survival cutoff. The estimation of hazard ratios (HR) was done by Cox proportional hazards model regression analysis. A 95% confidence interval was set and used. Patient samples with expression level above the threshold were considered as the high-expression or high-risk cohort. GEPIA2 was also used for CD73 and CD155 co-expression analysis and correlated using Pearson correlation.

GBM RNA-seq data from 156 patients were obtained through TCGA from the Genomic Data Commons. Linear correlation between normalized expression ·Fragments per Kilobase of transcript per Million mapped reads (FPKM) of target genes using corrplot R-package. Patients were classified into high/low groups for co-expression of CD155 (PVR) and CD73 (NT5E) using the upper and lower quartiles as cutoff for high and low expression, respectively. All codes for bioinformatic analysis used are publicly available at https://github.com/natallah/GBM_TCGA_analyses.

### Generation of engineered human iPSCs
Human iPS cell line hiPSC01 was thawed and cultured for ~5 days in 6-well plates to achieve ~50% confluency. Media was replaced with fresh media containing 5 ug/mL polybrene and lentiviral suspension

was added at an approximate multiplicity of infection (MOI) of 30. After 24 h this process was repeated. Following two rounds of transduction, cells were allowed to grow to ~80–85% confluency before being passaged. Cells were expanded and cryopreserved for future use.

### Differentiation of human iPSCs into NK cells and expansion
Engineered and non-engineered iPSCs were differentiated into NK cells utilizing our chemically-defined, serum- and feeder-free differentiation protocol, as previously described[37]. Briefly, hiPSC01 were thawed and cultured in mTesR™ 1 (StemCell Technologies, 85850) feeder-free maintenance medium for 4–5 days on hESC-Qualified Matrigel (Corning, 354277) coated 6-well plates to reach 80% confluency. iPSC lines were passaged using 0.5 mM EDTA in PBS (Biological Industries, 01-862-1B). Hematopoietic differentiation medium (HPDM) was prepared, consisting of StemDiff™ APEL™2 (StemCell Technologies, 05270), 40 ng/mL SCF (Shenandoah Biotechnologies), 20 ng/mL BMP4 (PeproTech, 120-05), and 20 ng/mL VEGF 165 (Shenandoah Biotechnologies), which was supplemented with 10 μM Rock inhibitor (Y-27632, Tocris, 1254) for the first 3 days. iPS cells were passaged and seeded in 100 μL of HPDM supplemented with Rock inhibitor in each well of an ultra-low attachment, round-bottom 96-well plate (Corning, CLS3474) at a density of 4500 cells/well. Cells were centrifuged for 5 min at 350 g to facilitate formation of embryoid body (EB) structures, and were incubated, undisturbed, for 3 days at 37 °C and 5% $CO_2$. On days 3, 6, and 9, media changes were performed by removing 70 μL of medium from each well and adding 100 μL of freshly prepared HPDM, without Rock inhibitor. On day 11, hematopoietic progenitor cells were either collected for flow cytometric analysis or transferred to NK cell differentiation cultures using a wide-bore p200 pipette (Fisher Scientific, 14-222-730).

Hematopoietic progenitor cells were seeded in a standard cell-culture treated 6-well plate at a concentration of 32 EBs per well in 3 mLs NK cell differentiation media (NKDM), consisting of StemDiff™ APEL™ 2 (StemCell Technologies, 05270), 20 ng/mL SCF (Shenandoah Biotechnologies), 20 ng/mL IL-7 (Shenandoah Biotechnologies), 10 ng/mL IL-15 (Shenandoah Biotechnologies) and 10 ng/mL Flt3L (Shenandoah Biotechnologies), and supplemented with 5 ng/mL IL-3 (Shenandoah Biotechnologies). Half media changes were performed twice per week for 4 weeks with freshly prepared NKDM containing IL-3 for the 1st week only, and without IL-3 for the following 3 weeks. After 4 weeks of NK cell differentiation culture, cells were collected and either analyzed phenotypically via flow cytometry or expanded for two to 3 weeks in CTS NK Xpander™ NK Cell Expansion medium (ThermoFisher) supplemented with 5% hAb serum (Corning, 35-060-CI), 1% penicillin/streptomycin (Gibco, 15140122), and 500 IU/mL rhIL-2 (Akron Biotech, AK8223). Cells were expanded in a 24 well G-Rex® plate (Wilson Wolf, 80192 M). Expansion was assessed via trypan blue staining.

### In vitro assessment of pharmacological blockade of TIGIT and CD73 on iNK cell killing of GBM43
Expanded iNK cells were cocultured with GBM43 WT cells for 4 h with or without anti-CD73 (clone 7G2, 10 ug/mL) and/or anti-TIGIT (clone A15153A, 50 ug/mL) blocking antibodies at effector:target (E:T) ratios of 2.5:1, 5:1, and 10:1 in CTS NK Xpander™ NK Cell Expansion complete

medium. Specific lysis was assessed by LDH activity using the CytoScan LDH assay kit (G Biosciences), following manufacturer protocol. In a separate experiment, TIGIT.SynNotch.aCD73 iNK cells along with WT iNK cells, with and without TIGIT and CD73 blockade, were similarly cocultured with GBM43 WT cells and cytolysis was assessed.

### Functional in vitro assessment of engineered iNK cells

Gene Expression and activation: Expanded engineered and non-engineered iNK cells were cultured with and without GBM43 WT or GBM43-CD155/CD73 KO target cells for 24 h to allow for activation, translation, and release of the anti-CD73 scFv in CTS NK Xpander™ NK Cell Expansion complete medium. After 48 h, cells were collected and measured for mCherry and EGFP expression via flow cytometry. GFP expression on mCherry + iNK cells indicates activation and translation of CD73 blocking scFv.

Phenotypic analysis: Immediately following differentiation, cells were stained and measured via flow cytometry for a panel of NK activating and inhibitory receptors, along with expression of the TIGIT.SynNotch genetic construct and intracellular GAL4 domain. Following expansion, cells were re-assessed for phenotypic marker expression.

Killing activity: Expanded iNK (WT and TIGIT.SynNotch.aCD73) cells were cocultured with GBM43 WT, GBM43-CD155 KO, GBM43-CD73 KO, GBM43-CD73/CD155 KO, GBM10, U87-MG, or SJ-GBM2 cells for 4 h at effector:target (E:T) ratios of 5:1 and 10:1 in CTS NK Xpander™ NK Cell Expansion complete medium. Specific lysis was assessed by LDH activity using the CytoScan LDH assay kit (G Biosciences), following manufacturer protocol.

Degranulation and cytokine production: iNK (WT and TIGIT.SynNotch.aCD73) cells were cocultured with GBM43 WT, for 4 h at effector:target (E:T) ratios of 5:1 and 10:1 in CTS NK Xpander™ NK Cell Expansion complete medium. CD107a-PE was added at the beginning of coculture. After 1 h, monensin (Biolegend) and brefeldin A (Biolegend) were added. After 4 h of culture, cells were washed and stained with CD56-APC, CD107a-PE, and IFNy-PerCP-Cy5.5 expression was assessed via flow cytometry.

iNK (WT and TIGIT.SynNotch.Acd73) cells were cocultured with GBM43 WT, for 24 h at effector:target (E:T) ratios of 5:1 and 10:1 in CTS NK Xpander™ NK Cell Expansion complete medium. Following coculture, supernatant was collected, and cytokine secretion was measured via ELISA (TNF-a, Invitrogen, TNF alpha Human ELISA Kit, KHC3011) according to manufacturer instructions.

### In vivo animal studies

All animal experiments described in this study, utilizing C57BL/6, NOD.Cg-Prkdc$^{scid}$ IL2rg$^{tm1Wjl}$/SzJ (NSG), and NOD.Cg-Prkdc$^{em26Cd52}$ IL-2$^{em26Cd22}$/NjuCrl (NCG) mice, were approved by the Purdue University Animal Care and Use Committee. Mice were euthanized according to the Purdue University Animal Care and Use Committee's guidelines for tumor growth (tumor volume > 2000 cm) and for signs of discomfort (weight loss, lethargy or hunch).

GL261 intracranial studies in immunocompetent mice: GL261 cells ($5 \times 10^5$) were stereotactically implanted in the right forebrain of sixteen 8 week old male C57BL/6 J (Strain#: 000664) via a burr hole created using a stereotactic delivery system at a depth of 3.5 mm. After 4 days, mice were randomly assigned to 4 experimental groups: PBS control ($n = 4$), anti-CD73 mAb (100 ug/mouse) ($n = 4$), anti-PVR mAb (100 ug/mouse) ($n = 4$), and anti-CD73 mAb (100 ug/mouse) and anti-PVR mAb (100 ug/mouse) ($n = 4$). Mice were treated biweekly for 3 weeks with mAbs or PBS by intra peritoneal (i.p.) injection. Following the study, mouse tumors were harvested for immunofluorescent analysis. Quantification of immunofluorescent analysis was performed on four random intratumoral sections from each treatment group at 100x magnification. For NKp46, CD3, FoxP3, LAG-3, PD-1, and GzB, stained cells were counted for each frame.

GL261 flank studies in immunocompetent mice: GL261 cells ($4 \times 10^6$) were inoculated subcutaneously (s.c.) in the right flank of twenty eight 6 week old male ($n = 14$) and female ($n = 14$) C57BL/6 J (Strain #: 000664) mice in collagen (LifeInk 200, Advanced Biosciences) diluted to 8.25 mg/mL in PBS. Tumor dimensions (L, W, H) and bodyweight measurements were monitored, and the tumor volume was calculated according to the formula: $V = 0.52 \times L \times W \times H$. After 20 days, tumors reached a volume of about 80 mm$^3$, and mice were randomly assigned to 4 experimental groups: PBS control ($n = 8$), anti-CD73 mAb (100 ug/mouse) ($n = 6$), anti-PVR mAb (100 ug/mouse) ($n = 6$), and anti-CD73 mAb (100 ug/mouse) and anti-PVR mAb (100 ug/mouse) ($n = 8$). Mice were treated biweekly for two and a half weeks with mAbs or PBS by intra peritoneal (i.p.) injection. Following treatment, blood, spleen, and tumors were harvested. Immune cells were isolated from blood, spleen, and tumors, and stained and analyzed via flow cytometry.

Intracranial xenograft model: Orthotopic patient-derived GBM xenograft models were generated using eighteen 8 week old male NSG (Strain #: 005557) mice as previously described[15]. Briefly, GBM43-Luc cells ($1 \times 10^5$) were stereotactically implanted in the right forebrain via a burr hole created using a stereotactic delivery system at a depth of 3.5 mm. After 9 days, mice were randomly assigned to three treatment groups: PBS ($n = 6$), WT iNK cells ($n = 6$), and TIGIT.SynNotch.aCD73 iNK cells ($n = 6$). Mice received weekly intracranial injections of 5 uL of either PBS or NK cells (suspended in 1% IgG-free BSA in PBS) for 3 weeks, and bioluminescence and bodyweight was measured biweekly until the mice reached a predetermined endpoint. D-Luciferin potassium salt was purchased from Syd Labs. Following the study, mouse tumors were harvested for histological analysis. Quantification of histological analysis was performed on four random intratumoral sections from each treatment group at 200x magnification. For NKp46 and Granzyme B, stained cells were counted for each frame. For CD73 and CD155, images were given a score for based on the following formula: 0 = 0% positive staining, 1 = 1–40% positive staining, 2 = 41–75% positive staining, 3 = 75–100% positive staining; 1 = weak staining, 2 = moderate staining, 3 = strong staining. The overall IHC score was calculated by multiplying each staining score. IHC array immunofluorescent staining was scored for each core using the same formula for both CD155 and CD73 expression. In a separate study, GBM43-Luc cells ($~8 \times 10^4$) were similarly stereotactically implanted in the right forebrain of five 8 week old male NSG mice. After 16 days, mice treated with intravenous (i.v.) injection of $5 \times 10^6$ primary NK cells. After 24 h, the mice were sacrificed and whole brain tissues from each mouse were harvested. The presence of NK cells was determined with histological (IHC) analyses by using anti-NKp46 antibody staining, as described.

GBM43 WT xenografts in immunodeficient mice: GBM43 WT cells ($3 \times 10^6$) were inoculated subcutaneously (s.c.) in the right flank of twenty five 6 week old male ($n = 15$) and female ($n = 10$) NCG (Strain #: 572) mice in collagen (LifeInk 200, Advanced Biosciences) diluted to 8.25 mg/mL in PBS. Tumor dimensions (L, W, H) and bodyweight measurements were monitored, and the tumor volume was calculated according to the formula: $V = 0.52 \times L \times W \times H$. After 14 days, tumors reached a volume of about 80 mm$^3$, and mice were randomly assigned to 5 experimental groups: PBS control ($n = 5$), WT iNK cells ($n = 5$), aCD73 iNK cells ($n = 5$), TIGIT.SynNotch iNK cells ($n = 5$), and TIGIT.SynNotch.aCD73 iNK cells ($n = 5$). Mice were treated weekly for 3 weeks with NK cells or PBS by intra venous (i.v.) injection with $5 \times 10^6$ NK cells per injection. Mice receiving NK cells also received 0.5 ug/injection rhIL-15 biweekly via intro peritoneal (i.p.) injection to support NK cell proliferation. Following treatment, tumors were harvested. NK cells were isolated from tumors and stained and analyzed via flow cytometry.

## Statistics and reproducibility

Prism 9 (Graphpad Software) was used for statistical analysis with a $p < 0.05$ considered significant. Ordinary one-way and two-way ANOVA tests along with Tukey's multiple comparison test or Sidak's multiple comparison test were used for multiple-group comparisons of unpaired samples. Unpaired two-tailed $t$-tests were used for single-data comparisons of independent groups. No statistical method was used to predetermine sample size. No data were excluded from analyses. Mice were randomized between treatment groups prior to treatment, and the investigators were not blinded to allocation during experiments and outcome assessment.

## Reporting summary

Further information on research design is available in the Nature Portfolio Reporting Summary linked to this article.

## Data availability

The open access TCGA-GBM (RNA-seq) publicly available data was used in this study are available via Genomics Data Commons (GDC) data portal (https://portal.gdc.cancer.gov/projects/TCGA-GBM). This data was accessed via R-Bioconductor package TCGAbiolinks (accessed April 2020). Source data for all figures are provided with this paper and are available online. Source TCGA data for Fig. 1A–C, Supplementary Fig. 1 was downloaded from the Gene Expression Profiling Interactive Analysis (GEPIA2) database and is available online. The authors declare that all data supporting the findings of the study are available in the article, the Supplementary file, and the Source Data provided with this paper. Source data are provided with this paper.

## Code availability

The complete code to access and analyze the TCGA-GBM data and to reproduce the Fig. 1G, H in the manuscript are available through this GitHub Repository (https://github.com/sagarutturkar/NatureComm_2024_Matosevic/).

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

## Acknowledgements

This work was supported by National Institutes of Health grant 1 R21CA256413-01 from the National Cancer Institute, the V Foundation for Cancer Research (Grant #D2019-039), the Walther Cancer Foundation (Embedding Tier I/II Grant #0186.01) and a McKeehan Graduate Fellowship to Kyle Lupo. The authors also gratefully acknowledge the support of the Collaborative Core for Cancer Bioinformatics, the Biological Evaluation Shared Resource and the Flow Cytometry Shared Resource, with support from the Purdue Center for Cancer Research, NIH grant P30 CA023168, the IU Simon Cancer Center NIH grant P30 CA082709, and the Walther Cancer Foundation. We also gratefully acknowledge Thermo Fisher Scientific (Gibco) for providing CTS™ NK-Xpander™ medium, and Wilson Wolf Corp. for providing G-Rex® plates.

## Author contributions

K.B.L. and S.M. conceived, designed and executed the experiments and wrote the manuscript; X.Y., S.B. and J.W. contributed to experimental work and manuscript editing; B.E., K.B.L., X.Y., S.B., J.W. and S.T-A. maintained mice and carried out in vivo xenograft studies; N.A.L. and S.U. performed bioinformatics analysis; M.M. performed histological analysis; all authors reviewed and edited the manuscript.

## Competing interests

The authors declare no competing interests.

## Additional information

**Kyle B. Lupo**[1], **Xue Yao**[1], **Shambhavi Borde**[1], **Jiao Wang**[1], **Sandra Torregrosa-Allen**[2], **Bennett D. Elzey**[2,3], **Sagar Utturkar**[2], **Nadia A. Lanman**[2,3], **MacKenzie McIntosh**[4] & **Sandro Matosevic** [1,2] ✉

[1]Department of Industrial and Molecular Pharmaceutics, Purdue University, West Lafayette, IN, USA. [2]Institute for Cancer Research, Purdue University, West Lafayette, IN, USA. [3]Department of Comparative Pathobiology, Purdue University, West Lafayette, IN, USA. [4]Histology Research Laboratory, Center for Comparative Translational Research, College of Veterinary Medicine, Purdue University, West Lafayette, IN, USA. ✉e-mail: sandro@purdue.edu

