## [Peer Review File · Nature Communications]

Reviewers' Comments:

Reviewer #1:

Remarks to the Author:

Immunotherapy for glioma has been largely unsuccessful so far. The authors tried to exploit the immunosuppressive ITIM domains (TIGIT) / CD155 axis for a novel therapeutic approach using adoptively transferred NK cells. They engineered induced pluripotent stem cell-derived NK (iNK) cells to allow for synNotch-mediated activation of a downstream signaling cascade, ultimately resulting in inhibition of CD73 function. Upon TIGIT/CD155 interaction, CD73 inhibition results in increased anti-glioma activity of iNK cells in vitro. In subcutaneous and intracranial xenograft models, local administration of the newly generated cells led to reduced tumor growth and prolonged survival.

Major comments:

The expression of CD155, TIGIT and CD73 has been described extensively in gliomas (e.g., Merrill et al. *Neuro Oncol* 2004; Sloan et al. *Cancer Res* 2005; Azambuja et al., *Cancer Immunol Immunother*; Gokhale et al. *J Immunother of Cancer* 2021 and many more). Therefore, the dataset shown in Fig. 1 adds only little novelty compared to the existing literature.

Data from most in vivo experiments is only partially convincing. It is not clear why the authors used several flank models in their experiments. All experiments shown in Fig. 4 rely on antibody administration and should have been done with an orthotopic glioma model. Tumor growth in the flank does not reflect the unique microenvironment in the CNS. While the presented data provides some insights in the effects of the applied antibodies, it does not allow drawing any conclusions on tumor growth in the brain as well as bystander cells in the CNS microenvironment upon adoptive transfer of NK cells.

In line with this, the xenograft models used for the next set of experiments come with many limitations. Again, data with flank models are of limited interest as gliomas are growing in the brain. More importantly, these models do not allow for an assessment of on-target/off-tumor toxicity. This aspect has not been explored at all in the manuscript. Any clinical translation would require a comprehensive preclinical assessment of the newly generated iNK cells. As all tools and antigen-binding sequences to engineer appropriate NK cells are basically available (Fig. 4), a full analysis in a syngeneic context should be feasible.

While intracranial treatment resulted in prolonged survival of glioma-bearing mice, this concept has many limitations. Did the authors assess the activity of their approach when iNK cells were administered systemically? Data of such experiments should be added to clarify if this strategy (that would be much more convenient in a clinical setting) results in anti-tumor activity in vivo. Importantly, it should be clarified to what extent immune escape mechanisms (e.g., loss of CD155 or CD73 expression) are observed under such conditions.

Minor points:

There is a lack of accuracy in the preparation of the reference list. Many identical citations are listed twice, e.g. refs. 13 and 18 or refs. 5 and 40. In line with this, there is a surprisingly high number of self-citations.

Information provided in the introduction and discussion sections is partially outdated, including a number of old reviews that are listed (e.g, refs. 1 and 27). In contrast, new information, e.g., on all clinical trials done with nivolumab in the glioblastoma field (e.g., Checkmate 143, 498, 548 trials) is lacking.

Reviewer #2:

Remarks to the Author:

The work by Kyle B. Lupo and colleagues investigates a novel therapeutic approach for GBM malignancies. The therapeutic approach revolves around hi-PS-derived natural killer cells, engineered to express an anti-CD155 synNotch that induces expression of anti-CD73 single-chain antibodies. The dataset collectively support a notion that double targeting of CD73 and CD155 is being beneficial for tumor containment.

The idea is innovative and some of the in vivo results are compelling (Fig. 5 and 6).

On the other hand, some of the in vivo results are a little puzzling and generate more questions than answers (Fig. 3 and 4); moreover, the in vitro data are described in a somewhat confusing manner (Fig. 1 and 2).

Overall, the work is valuable for a very deadly disease, so it's worth requesting revision.

Main points for revision.

a. Fig. 4 placement in the overall message is not too clear: if antibody treatment can be as good as cell treatment, why not do antibody treatment that could be perceived as safer? Are there reasons to believe that cell therapy is superior to antibody treatment? Could you do a direct comparison?

b. Many of the differences claimed, especially in vitro, are quite small, sometimes go in different directions, and sometimes the statistics are missing (see below). In my opinion there're 2 ways to deal with this: either do more experiments to achieve more statistical significance and more univocal message, or reduce the claimed understanding of mechanism, acknowledging that there could be other mechanistic explanations (either choices would be fine by me).

c. The effect on improving Nk exhaustion seems really mild and maybe not compatible with authors conclusion that double targeting inhibits other inhibitory functions of CD155.

Specific points.

Intro

Text. CD112 came out as a surprise for me at first read at the bottom of page 4

Fig. 1

Figure 1C correlation is very weak but the IHC and flow data is quite compelling in the other panels of the figure.

Fig. 1D, not clear where it come from; how are the samples chosen? Ordered? Its pretty pixelated...

Could not find description of how this specific IHC score was calculated for this figure.

Text describing Fig. 1D-E: it does not mention variability (big!) among the samples. Missed opportunity to introduce caution on finding interpretation and inherent variability in biological samples.

Fig. 1H: the text observes that there is co-expression in 50% of the tumors; said it that way it sounds like random: there's a 50% chance for a phenomenon, it's like coin tossing, no? Can you show some genes that do not correlate? How can you know that it is not just by chance? Any statistics?

Fig. 1I: data hard to interpret as I could not find information on what is the gray baseline?

Could not find supplementary figure captions; for characterization of CRISPR KO not clear what Fold MFI means, what is the sample, what is the error bars...

Fig. 2

Fig. 2A/B how is the genetic engineering performed? Lentivirus? At what stage of differentiation? Would be good to have this info in the main text.

Description of synNotch is rather summary; Adam17 is not the only protease; Fig. 2B is a schematic, of course, the text description need to reflect that this is the currently accepted model, not a proven reality.

Fig 2C does not support statement that "this shows that synNotch receptor is expressed"; data support mCherry expression; to support the claim (important!) should do staining of receptor independent from co-transcribed reporter; maybe with a tag? Would also be useful to distinguish it from the endogenous.

Not clear which cells are we looking at? It says in legend "iNK cells following differentiation"; what does that mean exactly?

Fig. 2C-E: not clear what the bar graphs are, and what are the outliers; how many times have the

experiment been repeated? There seems to be high variability in the outliers regardless; any comments on that?

Fig. 2E; it appears that there is a HUGE variability among samples in the iNK category; a reader may be confused as why that is, the authors should give some fair guidance on why they think it's the case.

Fig. 2J, is that different non -statistically-significant?

Fig. 2D-L are any of those differences statistically significant, or non-statistically-significant?

Fig.3:

Fig. 3A: the text claims are not really strongly supported by figures. And omission of comments on variability across ratio of cells.

Figure 3A, significance only for TIGET is shown, change of double treatment seems quite small especially at 10 W:T compared to CD73

Fig. 3B: not sure what is shown; not sure how many cells, how many replicates; how data are presented.

Fig. 3D-F are these differences non-statistically significant?

Fig. H, I, really hard to believe there is statistical significance, but may be just me. The differences in absolute values are minimal, it should be noted.

Figure 3C-I. The changes are quite small. This is probably due to the fact that the single KO and the double KO are already harder to kill also for WT Nk cells. In other words the presence of CD73 and CD155 already enhances the killing by NK cells anyway. One question: is the data from Fig. 3c through F done all together or on different experiments?

Page 9 "inability of engineered iNK cells to lyse targets without proper receptor/ligand activation" seems like an overstatement: the killing % are extremely similar in the 2 conditions.

Fig. 3J-M, also differences at different cell ratio, not mentioned; L/M differences are in different directions at different ratio - bizarre! In L the difference is statistically significant in the unstimulated state; why?

Fig. 3K, not sure where CD107a is shown; assuming it is on the x axis,

Figure 3J, L, M comparison should also be between all samples, particularly with unstimulated.

Figure 3K not cited in text?

Figure S7 no live/dead staining?

Fig. 4

strange figure story-wise, seems do not add too much to the storyline which is about iNK synNotch cells;

Fig. 4A-B, this is the first piece of data that I found very exciting!

Fig. 4D-I throughout characterization has lots of same issues with statistics as Fig. 3; not sure what to conclude from this dataset

Text: "indicating that dual targeting of CD73 and CD155 may enhance anti-tumor immune responses and mitigate functional exhaustion of effector immune cells, when compared to single targeting of CD155 or CD73 alone" is an overstatement based on the data; bars are moving up and down, very hard to make a supported claim with this dataset;

Could not find much comment in the text on Figures 4D-G

Figure 4H increase in infiltrating CD8 T cells and NK cells also happens with single antibodies so double treatment does not seem to be too beneficial. Same for CD4 t cells and M1 macrophages. For CD4 and CD8 the difference is also quite small. Also for all those data would be great to see all the datapoints in the graph as per previous figures. There seems to be change in neutrophils which is not noted. No stats on M2 macrophages.

Text: "Although the phenotype of tumor-infiltrating NK cells upon dual CD73/CD155 treatment was comparable to that of NK cells treated with CD155 alone, dual-treated tumors showed a significantly larger proportion of NK cells infiltrating the tumors" Hard to make out a concrete sense from this sentence; if the exhaustion phenotype (e.g. PD1 marker) between single treatment and double treatment is the same, but double has more NK cells, clearly NK cell number does not affect the exhaustion phenotype?

Fig. 5

Figure 5B, D, E data very compelling, but for this to become a therapy would be great to see a comparison of engineered cells vs antibody therapy to see what is best.

Fig. 5 D,E - strong!

F also strong; G not sure what I am looking at; percentage of what?

Figure 5F, G for PD1 the reduction is similar to synnotch only without aCD73, so double treatment not beneficial. Same for Tim3. Not sure what is the difference between F and G but some things are not consistent so if it's same analysis does not get same results.

Fig. 6

Figure 6 again would be great to compare with antibody treatment. Also without the single controls shown in figure 5 we cannot evaluate if the single targeting is best or worse.

Figure 6I, not really convinced by the images even though quantification seems striking.

Figure 6 K, N compelling

Reviewer #3:

Remarks to the Author:

This report further demonstrates the utility of the SynNotch system for enhancing the anti-tumor function of adoptive cell therapies.

The data are original and the pathways being targeted are relevant for cancer immunotherapy

There are questions of whether iPSC NK cells are comparable to primary NK cells in these adoptive transfer models however the ease of engineering iPSC-derived NK cells is a clear advantage

Response to Reviewers' Comments

Reviewer #1

Immunotherapy for glioma has been largely unsuccessful so far. The authors tried to exploit the immunosuppressive ITIM domains (TIGIT) / CD155 axis for a novel therapeutic approach using adoptively transferred NK cells. They engineered induced pluripotent stem cell-derived NK (iNK) cells to allow for synNotch-mediated activation of a downstream signaling cascade, ultimately resulting in inhibition of CD73 function. Upon TIGIT/CD155 interaction, CD73 inhibition results in increased anti-glioma activity of iNK cells in vitro. In subcutaneous and intracranial xenograft models, local administration of the newly generated cells led to reduced tumor growth and prolonged survival.

Major comments:

The expression of CD155, TIGIT and CD73 has been described extensively in gliomas (e.g., Merrill et al. Neuro Oncol 2004; Sloan et al. Cancer Res 2005; Azambuja et al., Cancer Immunol Immunother; Gokhale et al. J Immunther of Cancer 2021 and many more). Therefore, the dataset shown in Fig. 1 adds only little novelty compared to the existing literature.

Response: We thank the reviewer for this comment. To address it, we have provided additional new co-staining imaging data of patient brain tumor tissues to further highlight novelty in imaging. Though we agree that the expression of CD155, TIGIT, and CD73 has been described in gliomas, this has so far been done in independent studies, and the dual-targeting of CD155 and CD73 has yet to be described, and, therefore, provides novelty. We have updated our discussion and introduction to address this. We have also provided new data to show co-staining of CD73 and CD155 in human patient GBM tissue, the first time such an analysis has been reported.

See Figure 1, Introduction (Lines 54-55; 72-73), Discussion (Lines 345-374)

Data from most in vivo experiments is only partially convincing. It is not clear why the authors used several flank models in their experiments. All experiments shown in Fig. 4 rely on antibody administration and should have been done with an orthotopic glioma model. Tumor growth in the flank does not reflect the unique microenvironment in the CNS. While the presented data provides some insights in the effects of the applied antibodies, it does not allow drawing any conclusions on tumor growth in the brain as well as bystander cells in the CNS microenvironment upon adoptive transfer of NK cells.

Response: We thank the reviewer for this comment. This is an important point, and we have addressed it by providing new intracranial animal studies. These studies validate our flank models. With these new data, this and every major hypothesis made in our study is validated in an orthotopic brain cancer model. Specifically, we have collected new data in an orthotopic intracranial immunocompetent mouse model and have

confirmed the efficacy of dual targeting of CD155 and CD73 using targeted antibody therapy. We have also included fluorescent IHC showing infiltration of relevant immune populations within the tumor and tumor adjacent regions. By providing these data we show effects of our targeted therapy on immune cells in the brain tumor TME, as requested by the reviewer. These new data also validate our flank data and provide further confirmation of our therapeutic synNotch-based approach for brain tumors.

See Figure 4, S10-11 and Figure 4 caption

In line with this, the xenograft models used for the next set of experiments come with many limitations. Again, data with flank models are of limited interest as gliomas are growing in the brain. More importantly, these models do not allow for an assessment of on-target/off-tumor toxicity. This aspect has not been explored at all in the manuscript. Any clinical translation would require a comprehensive preclinical assessment of the newly generated iNK cells. As all tools and antigen-binding sequences to engineer appropriate NK cells are basically available (Fig. 4), a full analysis in a syngeneic context should be feasible.

Response: We thank the reviewer for this helpful comment and would like to clarify. Firstly, we agree flank models come with limitations and that is why we have supplemented them with new data using intracranial models. Specifically, we have collected new data in an orthotopic intracranial immunocompetent mouse model and have confirmed the efficacy of dual targeting of CD155 and CD73 using targeted antibody therapy (Figure 4). Additionally, we have established efficacy of our engineered iNK cell therapy in both flank and orthotopic intracranial xenograft models (Figure 5-6). To clarify, the use of flank studies was merely to screen our dual target iNK therapy against single targeted engineered iNK controls, given the difficult in establishing intracranial tumors in sufficient numbers of mice for rigorous comparison of all control groups (Figure 5). These approaches are common to pre-clinical brain therapy studies. With our new orthotopic data, we have validated every hypothesis in the manuscript in an orthotopic model, both immunocompetent and immunocompromised. We further demonstrated efficacy in an orthotopic intracranial model (Figure 6). Our orthotopic models are also unique as compared to many other GBM studies as they use patient-derived cells. In terms of availability of systems, targeting murine models requires murine iNK cells. To our knowledge, such murine differentiation has never been described before and no protocol for their derivation currently exists. Though we agree the appropriate antigen binding sequences are available, the protocols for generation of murine iNK cells are not available, and therefore, a full syngeneic model would be infeasible and outside the scope of the current study. The scope of our study was the development of human iNKs, which we have done here, for the first time in the literature for GBM therapy. To address on target, off tumor toxicities, we have including additional in vitro data showing that our engineered iNK cells do not specifically lyse a normal neuron cell line above WT iNK controls (Figure S8).

See Figures 4, S10-11, 5, 6, S8

While intracranial treatment resulted in prolonged survival of glioma-bearing mice, this concept has many limitations. Did the authors assess the activity of their approach when iNK cells were administered systemically? Data of such experiments should be added to clarify if this strategy (that would be much more convenient in a clinical setting) results in anti-tumor activity in vivo. Importantly, it should be clarified to what extent immune escape mechanisms (e.g., loss of CD155 or CD73 expression) are observed under such conditions.

Response: We thank the reviewer for this comment. Though we agree systemic administration of iNK cells would be more convenient, the current convention in GBM treatment with cell-based therapy has been *via* intratumoral infusions of cells, rather than i.v. infusions. Clinical (Brown CE et al., *N Engl J Med.*, 2016) and pre-clinical (Wang D et al., *Sci Transl Med.*, 2020) studies have all used intratumorally-injected CAR-T cells to treat GBM. Similarly, studies on CAR-NK cells have so far employed only local injections of these cells (Han J et al., *Sci Rep.*, 2015; Zhang C et al., *J Natl Cancer Inst.*, 2015; Genßler S et al., *Oncoimmunology*, 2015; Ma R et al., *Cancer Research*, 2021). Part of the reason has been the limited infiltration of systemically-infused immune cells into GBM brains, thus limiting their therapeutic efficacy. As we move toward clinical evaluation of this immunotherapy, we believe the initial convention will be to inject these cells intracranially into treated subjects. For that reason, our pre-clinical evaluation reported in this study represents a relevant proof-of-principle of a clinical treatment model.

Following the reviewer's suggestion, we have carried out an *in vivo* study to explore the homing of our engineered NK cells following their i.v. infusion in an orthotopic patient-derived GBM xenograft model. The detailed procedure of the study is illustrated below in Reviewer Only Figure 1A. Based on the results of the study, no obvious signal was detected in the brain of the mice under administration of CAR-NK cells alone, either in normal tissues or tumor tissues (Reviewer Only Figure 1B). In line with our prior experience with NK cell infusions and data reported in the manuscript, the relatively low signal observed upon i.v. administration may support the notion that superior responses to treatment could be achieved upon local NK cell delivery. Therefore, we focused on the local (intratumoral) injection of NK cells in the current study. Limited infiltration of NK cells from the systemic circulation into GBM has been discussed in the literature before (Burger MC et al., *Front Immunol.*, 2019).

Therefore, we believe intracranial administration most appropriately demonstrates the efficacy of our engineered iNK therapy. We have also already included IHC data showing changes in expression of CD73 and CD155 in our orthotopic intracranial model (Figure 6K-M) and have clarified the potential for immune escape mechanisms in the discussion.

See Figure 6, Reviewer Only Figure 1, Discussion

See Reviewer Only Figure 1 (shown below)

Reviewer Only Figure 1. Evaluation of NK cell homing following i.v. injection in an orthotopic patient-derived GBM xenograft model. (A) Schematic diagram showing the *in vivo* treatment program. Briefly, on day 0, the GBM43 cells ($\sim 8 \times 10^4$) were stereotactically implanted into the right forebrain through a burr hole created using a digitalized stereotactic delivery system. At specific days after implantation, mice were treated according to one of the following protocol: (1) intravenous (i.v.) injection of 5×10^6 multi-specific CAR-NK cells alone (suspended in $100 \mu\text{L}$ 1% BSA (IgG-free) in PBS); After 24 h, the mice were sacrificed and whole brain tissues from each mouse were harvested. Then, the presence of NK cells was determined with histological (IHC) analyses by using anti-NKp46 antibody staining. (B) Representative images of IHC staining for NK cells performed on brain sections from CAR-NK cells only treated mice. Among them, (1-3) stand for normal brain tissue areas. (4-5) stand for tumor areas. Scale bar = $25 \mu\text{m}$; $200\times$ magnification.

Minor points:

There is a lack of accuracy in the preparation of the reference list. Many identical citations are listed twice, e.g. refs. 13 and 18 or refs. 5 and 40. In line with this, there is a surprisingly high number of self-citations.

Response: We thank the reviewer for this comment. We have corrected these discrepancies. We have checked the references and removed duplicate citations, and have also removed some self-citations. The current citations only reflect work that is directly relevant to the current study or which represents foundational information/studies necessary to the work presented in this manuscript.

See References

Information provided in the introduction and discussion sections is partially outdated, including a number of old reviews that are listed (e.g. refs. 1 and 27). In contrast, new information, e.g., on all clinical trials done with nivolumab in the glioblastoma field (e.g., Checkmate 143, 498, 548 trials) is lacking.

Response: We thank the reviewer for this comment. We have updated this information and included relevant clinical trial data. We have also updated outdated reviews with more relevant ones published within the past year. We have also added information on recent Checkmate trials in GBM.

See Lines 47-49; See References

Reviewer #2

The work by Kyle B. Lupo and colleagues investigates a novel therapeutic approach for GBM malignancies. The therapeutic approach revolves around hi-PS-derived natural killer cells, engineered to express an anti-CD155 synNotch that induces expression of anti-CD73 single-chain antibodies. The dataset collectively support a notion that double targeting of CD73 and CD155 is being beneficial for tumor containment.

The idea is innovative and some of the in vivo results are compelling (Fig. 5 and 6).

On the other hand, some of the in vivo results are a little puzzling and generate more questions than answers (Fig. 3 and 4); moreover, the in vitro data are described in a somewhat confusing manner (Fig. 1 and 2).

Overall, the work is valuable for a very deadly disease, so it's worth requesting revision.

Main points for revision.

a. Fig. 4 placement in the overall message is not too clear: if antibody treatment can be as good as cell treatment, why not do antibody treatment that could be perceived as safer? Are there reasons to believe that cell therapy is superior to antibody treatment? Could you do a direct comparison?

Response: We thank the reviewer for this comment and thank them for the opportunity to clarify. Figure 4 is included to highlight the effect of dual targeting on NK cells and other immune cells in the context of a fully functional immune compartment. We believe an engineered iNK therapy is superior to antibody treatment due to poor infiltration of immune cells into the intracranial space and overall dysfunction and exhaustion of resident immune cells within GBM (Cozar et al, *Cancer Discov*, 2021). We have also included additional in vitro data to highlight that our engineered iNK cell therapy is superior to WT iNK cells with or without antibody therapy (Figure S9).

See Figure 4, S9

b. Many of the differences claimed, especially in vitro, are quite small, sometimes go in different directions, and sometimes the statistics are missing (see below). In my opinion there're 2 ways to deal with this: either do more experiments to achieve more statistical significance and more univocal message, or reduce the claimed understanding of mechanism, acknowledging that there could be other mechanistic explanations (either choices would be fine by me).

Response: We thank the reviewer for this comment. We have modified the figures to more appropriately highlight the differences observed between engineered and WT iNK cell therapies in vitro and have also modified the discussion to acknowledge that other mechanistic explanation may be at play (i.e. upregulation of CD73 expression and enzymatic capacity seen under hypoxic conditions within the TME, but not present in an in vitro coculture setting). Regarding statistical analysis, statistics were performed in all cases comparing all treatment groups within each study. For clarity of figures, only statistically significant comparisons were depicted, and we have clarified this in the figure captions.

See Figure 3, Figure Captions, Discussion

c. The effect on improving NK exhaustion seems really mild and maybe not compatible with authors conclusion that double targeting inhibits other inhibitory functions of CD155.

Response: We thank the reviewer for this comment. We have collected new data in an orthotopic intracranial immunocompetent mouse model and have confirmed the efficacy of dual targeting of CD155 and CD73 using targeted antibody therapy and to highlight the effect of dual targeting on NK cell activation and exhaustion. We have also included fluorescent IHC showing infiltration of relevant immune populations within the tumor and tumor adjacent regions. We would also like to highlight that CD155 has the capacity for NK cell activation through DNAM-1, inhibition through TIGIT, CD96 and KIR2DL5, as well as a tumor-intrinsic role. While inhibition through TIGIT is the dominant mechanism, we acknowledge the capacity of alternative mechanisms to be at play.

See Figure 4, S10-11

Specific points.

Intro

Text. CD112 came out as a surprise for me at first read at the bottom of page 4

Response: We thank the reviewer for this comment and have modified the introduction accordingly for clarity.

Fig. 1

Figure 1C correlation is very weak but the IHC and flow data is quite compelling in the other panels of the figure.

Fig. 1D, not clear where it come from; how are the samples chosen? Ordered? Its pretty pixelated...

Could not find description of how this specific IHC score was calculated for this figure.

Text describing Fig. 1D-E: it does not mention variability (big!) among the samples. Missed opportunity to introduce caution on finding interpretation and inherent variability in biological samples.

Fig. 1H: the text observes that there is co-expression in 50% of the tumors; said it that way it sounds like random: there's a 50% chance for a phenomenon, it's like coin tossing, no? Can you show some genes that do not correlate? How can you know that it is not just by chance? Any statistics?

Fig. 1I: data hard to interpret as I could not find information on what is the gray baseline?

Could not find supplementary figure captions; for characterization of CRISPR KO not clear what Fold MFI means, what is the sample, what is the error bars...

Response: We thank the reviewer for these helpful comments. In Figure 1C, this correlation is considered to be weak to moderate, and is significant, given that this is

human patient data. It additionally highlights that CD155 and CD73 expression are correlated only within GBM tumor regions and not in normal tissue.

Figure 1D depicts a brain tumor patient IHC array obtained from US Biomax, containing cores from low grade glioma (astrocytoma), GBM, and normal brain tissue. This has been replaced with IF staining of CD155 and CD73. The scoring is also specified in the methods section and is scored using the same method as the IHC depicted in Figure 6. We have updated the text to highlight variability and heterogeneity amongst GBM.

Figure 1H: We have updated the text to more clearly specify the correlation we observed. The association was determined based on the stratification of patients into CD73/CD155 high- and low-expressing groups. As such, the association is not entirely random because it has as a prerequisite baseline expression of one of these antigens. Our full, unstratified (raw) data includes genes that do not correlate, should this be of interest, and is available. The key point we are highlighting is that CD155 and CD73 expression can be correlated in a significant number of GBM patients and are therefore promising co-targets for engineered NK cell therapies.

Figure 1I: We thank the reviewer for this comment and have updated the figure caption to clarify that the gray baseline is an isotype control. We have also updated the figure caption for Figure S3 to clarify that the error bars indicate SD among three independent replicates and that fold MFI indicates fold change compared with an isotype control.

See Figure 1, Figure 1 caption, Table S1-2, Figure S3, Lines 118-122; 127-128

Fig. 2

Fig. 2A/B how is the genetic engineering performed? Lentivirus? At what stage of differentiation? Would be good to have this info in the main text.

Description of synNotch is rather summary; Adam17 is not the only protease; Fig. 2B is a schematic, of course, the text description need to reflect that this is the currently accepted model, not a proven reality.

Fig 2C does not support statement that “ this shows that synNotch receptor is expressed”; data support mCherry expression; to support the claim (important!) should do staining of receptor independent from co-transcribed reporter; maybe with a tag? Would also be useful to distinguish it from the endogenous.

Not clear which cells are we looking at? It says in legend “iNK cells following differentiation”; what does that mean exactly?

Fig. 2C-E: not clear what the bar graphs are, and what are the outliers; how many times have the experiment been repeated? There seems to be high variability in the outliers regardless; any comments on that?

Fig. 2E; it appears that there is a HUGE variability among samples in the iNK category; a reader may be confused as why that is, the authors should give some fair guidance on why they think it's the case.

Fig. 2J, is that different non -statistically-significant?

Fig. 2D-L are any of those differences statistically significant, or non-statistically-significant?

Response: We are thankful to the reviewer for these comments and opportunity to clarify. Figure 2A-B: We have outlined the genetic engineering in the methods section, but have also included a brief description in the results section for clarity. We have also outlined in the results section that other proteases can indeed mediated cleavage of our GALV9-VP64 transcription factor system.

Figure 2C: We have also included a figure showing the co-expression of mCherry and GAL4 protein in engineered primary NK cells (Figure S5). We have clarified this Figure and the terms in the text. These cells represent engineered iPSC-NK (iNK) cells after differentiation from iPSCs into mature NK cells. We have used this terms consistently throughout the manuscript.

Figure 2C-E: These figures depict flow cytometry phenotyping from 3 independent differentiations and the error bars indicate the SEM between these samples. We and others have previously described this variability amongst NK cells generated from iPSCs (Knorr et al, 2013, Stem Cells Transl Med; Zeng et al., 2017, Stem Cell Rep; Vitale et al., 2012, Stem Cells Transl Med; Lupo et al., 2021, Cytotherapy).

Figure 2J-L: This difference in expansion capacity is not statistically significant and error bars indicate SEM. We have updated the figure captions to clarify this. We have also have included an explanation of inherent variability amongst NK cells generated from iPSCs discussion section

Figure 2, Figure 2 caption, S5, See Results and Discussion Lines 156-161, 169-170

Fig.3

Fig. 3A: the text claims are not really strongly supported by figures. And omission of comments on variability across ratio of cells.

Figure 3A, significance only for TIGET is shown, change of double treatment seems quite small especially at 10 W:T compared to CD73

Fig. 3B: not sure what is shown; not sure how many cells, how many replicates; how data are presented.

Fig. 3D-F are these differences non-statistically significant?

Fig. H, I, really hard to believe there is statistical significance, but may be just me. The differences in absolute values are minimal, it should be noted.

Figure 3C-I. The changes are quite small. This is probably due to the fact that the single KO and the double KO are already harder to kill also for WT NK cells. In other words the presence of CD73 and CD155 already enhances the killing by NK cells anyway. One question: is the data from Fig. 3c through F done all together or on different experiments?

Page 9 "inability of engineered iNK cells to lyse targets without proper receptor/ligand activation" seems like an overstatement: the killing % are extremely similar in the 2 conditions.

Fig. 3J-M, also differences at different cell ratio, not mentioned; L/M differences are in different directions at different ratio - bizarre! In L the difference is statistically significant in the unstimulated state; why?

Fig. 3K, not sure where CD107a is shown; assuming it is on the x axis,

Figure 3J, L, M comparison should also be between all samples, particularly with unstimulated.

Figure 3K not cited in text?

Figure S7 no live/dead staining?

Response: We thank the reviewer for these helpful comments. We have updated the claims to more closely describe the figures presented and to address variability across different E:T ratios. We would also like to highlight that these data are for antibody treatment, not for engineered iNK cells. In the data, no significant difference from the WT control was observed with either single antibody control, and only with dual antibody blockade was there a significant improvement in tumor lysis, demonstrating the advantage of dual targeting of CD73 and CD155 (Figure 3A).

Figure 3B: These data were collected via flow cytometry following 24 hour coculture with CD73+/CD155+ and CD73-CD155- GBM43 tumor targets, as described in the methods. Depicted are the percentage of activated, CD73 scFv-secreting cells (measured by a GFP reporter) gated on synNotch positive iNK cells (measured by an mCherry reporter). We have updated the methods and figure captions to clarify this.

Figure 3D-F: No statistical significance was observed here. As consistent throughout the manuscript, only statistically significant comparisons were reports within the figure for clarity. We have updated the methods and figure captions to clarify this. **Figure C-I:** We have updated these figures to more clearly show differences between WT and engineered iNK groups. These experiments were done in three independent replicates from iNK cells generated from three independent differentiations. We have also updated the text to subdue the claims in line with the reviewer's comments.

Figure 3J-M: We have updated the text and figures to clarify differences at different E:T ratios. We have compared different treatment groups at each E:T ratio, as well as unstimulated samples between each treatment group, and have reported only statistically significant comparisons, consistent with the rest of the manuscript. We did not compare samples across E:T ratios or with unstimulated controls, as we are interested in showing differences associated with treatment, as is standard for these types of assays. Differences between unstimulated cells can be explained by inherent variability in differentiation of iNK cells and we have addressed this in the results. We have cited Figure 3K in the text and updated this figure with appropriate axes labels. **Figure S10:** We have updated this figure to include the live/dead staining panel which was inadvertently omitted.

Figure 3, S12, See Lines 182-184, 569-570, 202-209, 212, 389-391, 647-648

Fig. 4

strange figure story-wise, seems do not add too much to the storyline which is about iNK synNotch cells;

Fig. 4A-B, this is the first piece of data that I found very exciting!

Fig. 4D-I throughout characterization has lots of same issues with statistics as Fig. 3; not sure what to conclude from this dataset

[SEP]Text: "indicating that dual targeting of CD73 and CD155 may enhance anti-tumor immune responses and mitigate functional exhaustion of effector immune cells, when compared to single targeting of CD155 or CD73 alone" is an overstatement based on the data; bars are moving up

and down, very hard to make a supported claim with this dataset;

Could not find much comment in the text on Figures 4D-G

Figure 4H increase in infiltrating CD8 T cells and NK cells also happens with single antibodies so double treatment does not seem to be too beneficial. Same for CD4 t cells and M1 macrophages. For CD4 and CD8 the difference is also quite small. Also for all those data would be great to see all the datapoints in the graph as per previous figures. There seems to be change in neutrophils which is not noted. No stats on M2 macrophages.

Text: "Although the phenotype of tumor-infiltrating NK cells upon dual CD73/CD155 treatment was comparable to that of NK cells treated with CD155 alone, dual-treated tumors showed a significantly larger proportion of NK cells infiltrating the tumors" Hard to make out a concrete sense from this sentence; if the exhaustion phenotype (e.g. PD1 marker) between single treatment and double treatment is the same, but double has more NK cells, clearly NK cell number does not affect the exhaustion phenotype?

Response: We thank the reviewer for this comment and would like to clarify. Figure 4 is included to highlight the effect of dual targeting on NK cells and other immune cells in the context of a fully functional immune compartment. Since these co-targets represent a novel combination therapy, we believe showing the effect of their targeting on the immune environment in GBM is worthwhile and adds context to our overall study.

We have tempered the statement as suggested above, and have rather highlighted the effects of the dual blockade on tumor control and immune infiltration.

See Lines 245-251

Figure 4D-I: As stated, statistics were calculated comparing all treatments groups to all other treatment groups, and, for clarity of figures, only statistically significant comparisons are depicted. We have clarified this in the methods and figure captions. We have also updated the discussion to subdue the claims and describe this dataset in more detail in line with the reviewer's comments. Additionally, Figures 4D-I are now Figures 4 I-J, S11-13 and are mentioned on Lines 230-243

See Figure 4; See Lines 230-243, 370-374

Figure 4H: We agree with the reviewer, that although some phenotypic comparisons are similar between single and dual antibody treatments, the improved tumor control coupled shows a clear advantage in dual targeting of CD73 and CD155 over single targeting of either ligand. In addition, however, our data point to the fact that phenotype alone does not define NK cell dysfunction in GBM, which contributes novel information to our understanding of NK cell exhaustion. We have also collected new data in an orthotopic intracranial immunocompetent mouse model and have confirmed the efficacy of dual targeting of CD155 and CD73 using targeted antibody therapy. We have also included fluorescent IHC showing infiltration of relevant immune populations within the tumor and tumor adjacent regions.

Figure 4, S10-11; See Lines 250-251

Fig. 5

Figure 5B, D, E data very compelling, but for this to become a therapy would be great to see a

comparison of engineered cells vs antibody therapy to see what is best.

Fig. 5 D,E - strong!

F also strong; G not sure what I am looking at; percentage of what?

Figure 5F, G for PD1 the reduction is similar to synnotch only without aCD73, so double treatment not beneficial. Same for Tim3. Not sure what is the difference between F and G but some things are not consistent so if it's same analysis does not get same results.

Response: We thank the reviewer for these comments. We have included additional in vitro data showing superiority of our engineered iNK therapy over antibody treatment (Figure S9).

Figures 5F-G depict fold increase in MFI over an isotype control and percentage of tumor infiltrating NK cells expressing each surface ligand, respectively. We have updated these figure captions to clarify this. Though we agree, TIGIT.SynNotch and TIGIT.SynNotch.aCD73 iNK cells express similar levels of PD-1 and TIM-3, these data highlight that TIGIT blockade can mitigate exhaustion compared with WT iNK controls. However, single targeting of TIGIT, though mitigating upregulation of exhaustion markers, is not sufficient to yield a response in tumor control (Figures 6B-E). Therefore, dual targeting of TIGIT and CD73 is necessary to mediate a robust anti-tumor response. We have updated the discussion to clarify this.

Figure 5 and Figure 5 caption, S9, Discussion (Lines 345-374)

Fig. 6

Figure 6 again would be great to compare with antibody treatment. Also without the single controls shown in figure 5 we cannot evaluate if the single targeting is best or worse.

Figure 6I, not really convinced by the images even though quantification seems striking.

Figure 6 K, N compelling

Response: We thank the reviewer for these helpful comments. We have included additional in vitro data showing superiority of our engineered iNK therapy over antibody treatment (Figure S9). To clarify, we utilized flank studies to screen our dual target iNK therapy against single targeted engineered iNK controls, given the difficulty in establishing intracranial tumors in sufficient numbers of mice for rigorous comparison of all control groups (Figure 5). We have clearly demonstrated in this flank model that dual targeting is superior to single targeting, and therefore, because of the limitations and challenges related to performing intracranial do not believe it is necessary to compare these inferior control groups in our orthotopic model. We have also updated Figure 6I with new IHC images to more clearly demonstrate differences observed in NK cell infiltration between our treatment groups.

Figure 5, 6, S9

Reviewer #3

This report further demonstrates the utility of the SynNotch system for enhancing the anti-tumor function of adoptive cell therapies.

The data are original and the pathways being targeted are relevant for cancer immunotherapy

There are questions of whether iPSC NK cells are comparable to primary NK cells in these adoptive transfer models however the ease of engineering iPSC-derived NK cells is a clear advantage

Response: We thank the reviewer for their comments.

Reviewers' Comments:

Reviewer #1:

Remarks to the Author:

The authors provide a revised manuscript with additional data. However, several limitations remain, such as (i) the limited novelty of expression data in Fig. 1, (ii) the excessive use of flank models, which, contrary to the authors' statement are not "common to pre-clinical brain therapy studies" and (iii) the new Kaplan-Meier plot shown in Fig.6, which is prematurely stopped at day 45. Longer observation is needed to come to a meaningful conclusion. Finally, having "reviewer only figures", which are not part of the manuscript, is discouraged as the fact that the approach is inactive when systemic administration is used, represents a major limitation that should not be withheld from the scientific community.

Reviewer #2:

Remarks to the Author:

The revision of the work put forward by Lupo et al., has addressed many of the points raised by me and the other reviewers. The manuscript present novel data in a rigorous manner. The revision added clarity to the presentation of the results, and also added some relevant in vivo dataset that support authors claims. I only have a minor comment (see below) at this point.

One minor comment could be add regarding synNotch activation: if I understand correctly, from fig. 3B it seems like the % of cells that produce GFP is 5%, and, upon induction of synNotch, becomes 10%, which is relatively low % compared to other synNotch-based circuits in primary T-cells. In contrast, the in vivo results are very strongly supporting that these differences make a lot of difference in vivo in terms of tumor survival (Fig. 6B for example). Can the author speculate why they think it is the case that a difference of just 5% in cell activation makes a big difference in terms of tumor clearance? Maybe something worth discussing in the discussion of the results.

Response to the reviewer comments

Reviewer 1

The authors provide a revised manuscript with additional data. However, several limitations remain, such as (i) the limited novelty of expression data in Fig. 1, (ii) the excessive use of flank models, which, contrary to the authors' statement are not "common to pre-clinical brain therapy studies" and (iii) the new Kaplan-Meier plot shown in Fig.6, which is prematurely stopped at day 45. Longer observation is needed to come to a meaningful conclusion. Finally, having "reviewer only figures", which are not part of the manuscript, is discouraged as the fact that the approach is inactive when systemic administration is used, represents a major limitation that should not be withheld from the scientific community.

Response: We have addressed the above comments. We have focused on orthotopic models, by including them in the manuscript and moving flank models to the supplementary, have extended survival data, and have included the "ref only figure" as a supplementary figure.

See Figures 4, 5, 6 and Supplementary Figure 16.

Reviewer #2

The revision of the work put forward by Lupo et al., has addressed many of the points raised by me and the other reviewers. The manuscript present novel data in a rigorous manner. The revision added clarity to the presentation of the results, and also added some relevant in vivo dataset that support authors claims. I only have a minor comment (see below) at this point.

One minor comment could be add regarding synNotch activation: if I understand correctly, from fig. 3B it seems like the % of cells that produce GFP is 5%, and, upon induction of synNotch, becomes 10%, which is relatively low % compared to other synNotch-based circuits in primary T-cells. In contrast, the in vivo results are very strongly supporting that these differences make a lot of difference in vivo in terms of tumor survival (Fig. 6B for example). Can the author speculate why they think it is the case that a difference of just 5% in cell activation makes a big difference in terms of tumor clearance? Maybe something worth discussing in the discussion of the results.

Response: We have addressed the above comment. We have included a brief discussion about the difference in activation and its impact on tumor clearance. In doing so, we hypothesize that secreted aCD73 scFv from even a small percentage of synNotch-activated iNK cells, may be blocking the inhibitory effects of CD73, even for non-synNotch-activated iNK cells as a bystander effect, thus enhancing the anti-tumor response in vivo.

See Lines 352-362